# A Survey on Self-Supervised Representation Learning

## Abstract

Learning meaningful representations is at the heart of many tasks in the field of modern machine learning. Recently, a lot of methods were introduced that allow learning of image representations without supervision. These representations can then be used in downstream tasks like classification or object detection. The quality of these representations is close to supervised learning, while no labeled images are needed. This survey paper provides a comprehensive review of these methods in a unified notation, points out similarities and differences of these methods, and proposes a taxonomy which sets these methods in relation to each other. Furthermore, our survey summarizes the most-recent experimental results reported in the literature in form of a meta-study. Our survey is intended as a starting point for researchers and practitioners who want to dive into the field of representation learning.

## 1 Introduction

Images often contain a lot of information which is irrelevant for further downstream tasks. A *representation* of an image ideally extracts the relevant parts of it. The goal of *representation learning* is to learn an *encoder network* $f_\theta$ with learnable parameters $\theta$ that maps an input image $x$ to a lower-dimensional representation (embedding) $y = f_\theta(x)$. The main purpose of this paper is to present and discuss the different approaches and ideas for finding useful encoders. Note that in the machine learning literature this setup is also called feature learning or feature extraction.

### 1.1 Supervised and Unsupervised Learning

Before categorizing the various methods of representation learning, we begin by distinguishing the two basic settings of machine learning: (i) *supervised learning* learns a function given labeled data points, interpreted as input and output examples, (ii) whereas in *unsupervised learning* we learn something given only data points (unlabeled). It is possible to take the view that also for unsupervised learning the goal is to learn a function, e.g., in clustering we learn a classification function and invent labels at the same time, in dimensionality reduction (such as PCA, ISOMAP, LLE, etc.) we learn a regression function and invent lower-dimensional embeddings simultaneously. In light of this, representation learning is an instance of unsupervised learning, since we are only given a set of unlabeled data points and our goal is to learn an encoder, that maps the data onto some other representation that has favorable properties.

### 1.2 Self-Supervised Learning

Recently, new machine learning methods emerged, that have been labeled *self-supervised learning*. In a nutshell, such methods create an encoder by performing the following two separate steps:

(i) Invent a supervised learning task by creating a output $t$ for each given image $x$.

(ii) Apply supervised learning to learn a function from inputs $x$ to targets $t$.

The mechanism that generates the targets can be manually designed or can include learned neural networks. Note that the targets are not necessary stationary, but could change during training. Even though self-

supervised learning applies classical supervised learning as its second step, it is overall best viewed as an unsupervised method, since it only takes unlabeled images as its starting point.

### 1.3 Outline

This paper strives to give an overview over recent advances in representation learning. Starting from the autoencoder, we will discuss different methods which we group in the following way:

- *Pretext task methods* solve an auxiliary task, e.g., predicting the angle by which an input image was rotated. The idea is that the representations learned along the way are also useful for solving other tasks. To create a pretext task, one usually needs to select image transformations and define a new loss function. The encoder will only extract information that is relevant for solving this task, so extra care must be taken. We will further discuss these methods in Section 2.

- *Information maximization methods* allow networks to learn representations that are invariant to various image transformations and at the same time avoid trivial solutions by maximizing information content. These representations prove useful for other downstream tasks, such as classification. Much like with pretext task methods, the transformations need to be chosen with care, as they influence which information is targeted to be preserved or suppressed. Section 3 presents some of these methods in detail.

- *Teacher-student methods* consist of two networks where one extracts knowledge from the other. Trivial solutions are avoided by using asymmetric architectures where the teacher provides stable representations for the student to learn, while invariances to various image transformations are usually distilled onto the teacher. We will take a closer look at these methods in Section 4.

- *Contrastive methods* discriminate between positive and negative examples that are defined by the method on-the-fly. Maximizing the similarity between positive examples while repelling negatives offers a powerful way to extract rich and context-specific features. However, selecting large numbers of appropriate negative examples is a challenging task. In Section 5 we review these contrastive methods in detail.

- *Clustering-based methods* invent multiple class labels by clustering the representations and then train a classifier on those labels. Intuitively, semantically similar images should lead to similar representations that should also be assigned to the same clusters by some clustering approach. Cluster assignments can replace memory inefficient feature comparisons of negative samples from memory banks and also provide some interpretability of the image representations. However, the lack of labels in the unsupervised environment require more sophisticated training in this setup. Section 6 summarizes the most relevant representation learning methods that use clustering.

Note that the order in which we present the methods in this paper is not the same as the order in which they have been introduced in the literature.

In Section 7 we further put the discussed methods in relation to each other and in Section 8 we summarize experimental results that were reported in the literature in a meta study.

While we focus on approaches for visual representation learning in this paper, there exist further approaches which are specialized for graphs (Grover & Leskovec, 2016; Perozzi et al., 2014; Kipf & Welling, 2016), time-series (Eldele et al., 2021) or text (Mikolov et al., 2013b;a; Kenton & Toutanova, 2019), which we omit.

### 1.4 Notation

Before describing the specifics of the different representation learning methods, we start by defining the notation used in this paper. Given a dataset of images, we write

$$X = [x_1, \ldots, x_n] \tag{1}$$

for a randomly sampled batch of images. Every representation learning method trains an encoder network $f_\theta$, where $\theta$ are the learnable parameters. This encoder network computes a representation

$$Y = [y_1, \ldots, y_n] = [f_\theta(x_1), \ldots, f_\theta(x_n)] = f_\theta(X) \tag{2}$$

of the images in $X$. Some methods additionally train a projection network $g_\phi$, with parameters $\phi$, that computes projections

$$Z = [z_1, \ldots, z_n] = [g_\phi(y_1), \ldots, g_\phi(y_n)] = g_\phi(Y) \tag{3}$$

of the representations in $Y$. There are methods that also train a prediction network $q_\psi$, with parameters $\psi$, that computes a prediction based on $z$ or $y$. Both, projections and predictions are only used to train the network and after training the projection and prediction networks are discarded, and only the encoder $f_\theta$ is used for downstream tasks.

Most methods apply a transformation to the input image to obtain a *view* of the original image. We write $x_i^{(j)} = t(x_i)$ for the $j$-th view of the original image $x_i$, which was obtained by applying the transformation $t$. Usually, these methods randomly sample transformations from a given set of transformations $\mathcal{T}$ and can differ for each image in the batch. This is why we treat $t$ as a random variable that is sampled for each image of the batch. In contrast to that, other methods use predefined transformations $t^{(1)}, \ldots, t^{(m)}$ that are fixed and do not change. In Section 3 we give more details and some examples of these transformations. We write

$$X^{(j)} = [x_1^{(j)}, \ldots, x_n^{(j)}] = t([x_1, \ldots, x_n]) = t(X) \tag{4}$$

for the batch of the $j$-th view. We use a similar notation for the resulting representations $Y^{(j)} = [y_1^{(j)}, \ldots, y_n^{(j)}]$ and projections $Z^{(j)} = [z_1^{(j)}, \ldots, z_n^{(j)}]$. Some methods split the input image into patches, which we also consider as a special case of transformation, where each patch is a view of the image. In that case we write $X_i = [x_i^{(1)}, \ldots, x_i^{(m)}]$, which contains all $m$ views of the image $x_i$ and denote the corresponding representations and projections as $Y_i = [y_i^{(1)}, \ldots, y_i^{(m)}]$ and $Z_i = [z_i^{(1)}, \ldots, z_i^{(m)}]$.

In some cases, the calculation of the representations can be decoupled such that each view is processed independently, i.e., $Y_i = [f_\theta(x_i^{(1)}), \ldots, f_\theta(x_i^{(m)})] = f_\theta(X_i)$. If this is possible we call the corresponding encoder a *Siamese* encoder. The same distinction can be made for the other networks (projector and predictor). A Siamese projector computes $Z_i = [g_\phi(y_i^{(1)}), \ldots, g_\phi(y_i^{(m)})] = g_\phi(Y_i)$ individually. However, as we will see later, this is not always the case as some networks operate on multiple views simultaneously.

We use $\mathcal{L}$ to refer to the loss function that is used to train the parameters using stochastic gradient descent (SGD). Sometimes the total loss consists of multiple components which we denote using the letter $\ell$.

We use square brackets to access elements of vectors and matrices, e.g., the $j$-th element of vector $v$ is written as $v[j]$ and the entry at column $j$ and row $k$ of matrix $C$ is written as $C[j, k]$. Furthermore, we define the softmax function that normalizes a vector $v \in \mathbb{R}^d$ to a probability distribution as

$$(\text{softmax}_\tau(v))[j] = \frac{\exp(v[j]/\tau)}{\sum_{k=1}^d \exp(v[k]/\tau)} \text{ for } j = 1, \ldots, d, \tag{5}$$

where $\tau > 0$ is a temperature parameter which controls the entropy of that distribution (Wu et al., 2018). The higher the value of $\tau$ the closer the normalized distribution is to a uniform distribution. We write $\text{softmax}(v)$ when no temperature is used, i.e., when $\tau = 1$.

**Distance metrics and similarity measures.** Throughout the paper, we use various concepts to compare two vectors. We introduce these concepts in the following. First, we define the squared error between two vectors $v, w \in \mathbb{R}^d$ as

$$d_{\text{se}}(v, w) = \|v - w\|_2^2 = (v - w)^\top (v - w) = \sum_{j=1}^d (v[j] - w[j])^2. \tag{6}$$

Note that the squared error is the same as the squared Euclidean norm of the residuals. Sometimes, the vectors $v$ and $w$ are normalized before the squared error is calculated. We denote the resulting distance metric as normalized squared error

$$d_{\text{nse}}(v, w) = \left\| \frac{v}{\|v\|_2} - \frac{w}{\|w\|_2} \right\|_2^2. \tag{7}$$

We define the cosine similarity as

$$s_{\text{cos}}(v, w) = \frac{v^\top w}{\|v\|_2 \|w\|_2}. \tag{8}$$

Note that $d_{\text{nse}}$ and $s_{\text{cos}}$ are linearly related, i.e.,

$$d_{\text{nse}}(v, w) = 2 - 2 \frac{v^\top w}{\|v\|_2 \|w\|_2} = 2 - 2 s_{\text{cos}}(v, w) \tag{9}$$

$$\Leftrightarrow s_{\text{cos}}(v, w) = 1 - \frac{1}{2} d_{\text{nse}}(v, w). \tag{10}$$

To measure the distance between two discrete probability distributions described by the entries of two vectors $v$ and $w$, the cross-entropy can be used which is given as

$$d_{\text{ce}}(v, w) = -\sum_{j=1}^{d} v[j] \log w[j]. \tag{11}$$

Note that the cross-entropy does not fulfill the triangle inequality and is therefore no distance metric in a mathematical sense. A common loss function for multiclass classification is the cross-entropy between scores $v \in \mathbb{R}^d$ that are normalized using softmax and a one-hot distribution of the true class label $c \in \mathbb{N}$, which we denote by

$$d_{\text{classification}}(v, c) = d_{\text{ce}}(\text{onehot}(c), \text{softmax}(v)) \tag{12}$$

$$= -v[c] + \log \left( \sum_{j=1}^{d} \exp v[j] \right), \tag{13}$$

where $\text{onehot}(c)$ is a vector with 1 in the $c$-th component and 0 everywhere else.

## 2 Pretext Task Methods

In the introduction we have defined the concept of self-supervised learning which relies on defining a mechanism that creates targets for a supervised learning task. There are many possibilities to invent such supervised learning problems. These learning problems are usually called *pretext* tasks. The idea is that the features learned by solving the pretext task are also helpful for solving other problems. In the following we present works, that creatively came up with such tasks.

### 2.1 Autoencoders

Autoencoders (Le Cun, 1987) have been part of machine learning for a very long time and in the light of the previous section, they can be seen as early instances of self-supervised learning: (i) the invented targets are the inputs themselves and (ii) the learned function is a bottleneck neural network consisting of an encoder $f_\theta$ that maps an image $x_i$ to a low-dimensional representation $y_i = f_\theta(x_i)$ and a predictor $q_\psi$ that reconstructs the input image $\hat{x}_i = q_\psi(y_i)$ from its representation.

Given a batch of images $X$, the encoder and predictor networks are jointly trained to minimize the reconstruction error

$$\mathcal{L}_{\theta,\psi}^{\text{AE}} = \frac{1}{n} \sum_{i=1}^{n} d_{\text{se}}(\hat{x}_i, x_i). \tag{14}$$

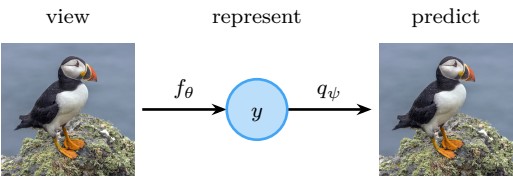

Figure 1: An autoencoder consists of two networks: an encoder $f_\theta$ that maps the input image to a representation and a predictor $q_\psi$ that is trained to reconstruct the original image from the representation.

There are many variants of the autoencoder: denoising autoencoder (Le Cun, 1987), stacked denoising autoencoder (Vincent et al., 2010), contractive autoencoder (Rifai et al., 2011) or variational autoencoder (VAE, Kingma & Welling, 2013).

## 2.2 Predicting Image Rotations (RotNet)

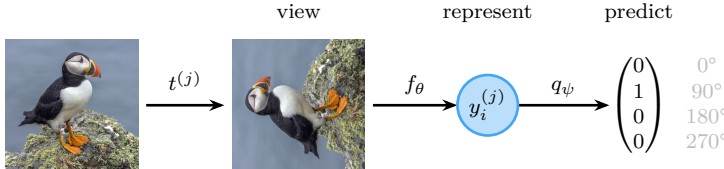

Figure 2: RotNet solves the task of predicting image rotations to obtain representations for downstream tasks.

The idea of the RotNet which was introduced by Gidaris et al. (2018) is to learn a representation that is useful to predict the angle of a random rotation applied to the input image. The assumption is that a representation that can predict the rotation is also valuable for other tasks. The authors show that even a small number of rotations is sufficient to learn a good representation. Best results are obtained when the number of rotations is four (0°, 90°, 180°, 270°). In that case the rotation augmentation can be efficiently implemented using flips and transposes and no interpolation is needed.

Given a batch of images $X$, we consider a single image $x_i$. Four views $x_i^{(j)} = t^{(j)}(x_i)$ are created using the rotation transformations $t^{(1)}, t^{(2)}, t^{(3)}, t^{(4)}$. The Siamese encoder $f_\theta$ converts each view into a representation $y_i^{(j)} = f_\theta(x_i^{(j)})$. The Siamese predictor $q_\psi$ is then used to predict the index of the rotation that was applied to the original image. Both networks are trained by minimizing the classification loss

$$\mathcal{L}_{\theta,\psi}^{\text{RotNet}} = \frac{1}{n} \sum_{i=1}^{n} \sum_{c=1}^{4} d_{\text{classification}}(q_\psi(y_i^{(c)}), c) \tag{15}$$

for each of the four rotations. After training, the classification head $q_\psi$ is discarded and only $f_\theta$ is used to calculate representations (of images that were not rotated). The authors use a Network-in-Network architecture (Lin et al., 2013) for their experiments on CIFAR-10 and the AlexNet architecture (Krizhevsky et al., 2017) for experiments on ImageNet.

## 2.3 Solving Jigsaw Puzzles

Jigsaw (Noroozi & Favaro, 2016) is similar to RotNet in the sense that it also solves a classification task. However, instead of rotating the image, the transformation consists of randomly permuting several patches of the image like a jigsaw puzzle. The pretext task of the model is then to predict the class of the permutation that was used to shuffle the patches. To facilitate the task, it is necessary to restrict the used permutations to a subset of all permutations. In their work the authors use 1000 predefined permutations (instead of 9! = 362 880 for a 3 × 3 grid of tiles).

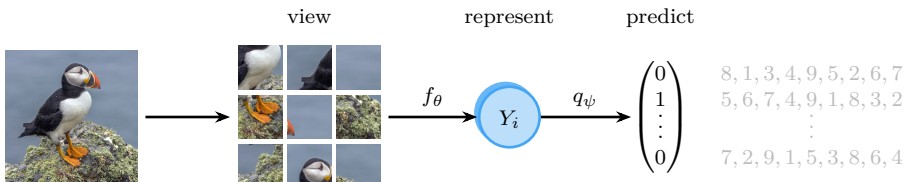

Figure 3: Jigsaw extracts patches from an image that are then permuted. The pretext task is to find the permutation that was used to permute the images. The context-free network $f_\theta$ processes each patch separately and the representations are only joined in the latter layers.

From an input image $x_i$ nine non-overlapping randomly permuted patches $[x_i^{(1)}, \ldots, x_i^{(9)}]$ are extracted, where the order of the patches follows one of the predefined permutations. After that, the Siamese encoder $f_\theta$ converts each patch into a representation $y_i^{(j)} = f(x_i^{(j)})$. The predictor $q_\psi$ is used to predict the index $c_i$ of the permutation that was applied to the original image, given all patch representations $Y_i = [y_i^{(1)}, \ldots, y_i^{(9)}]$ at once. The networks are trained by minimizing the loss

$$\mathcal{L}_{\theta,\psi}^{\text{Jigsaw}} = \frac{1}{n} \sum_{i=1}^{n} d_{\text{classification}}(q_\psi(Y_i), c_i) \tag{16}$$

between the class scores and the index of the used permutation $c_i$. The encoder $f_\theta$ is implemented as a truncated AlexNet. The representations $Y_i$ are concatenated to form the input of the classification head $q_\psi$, which is implemented as a multi-layer perceptron (MLP). After training, the classification head is discarded and the encoder is used to obtain image representations for other downstream task.

## 2.4 Masked Autoencoders (MAE)

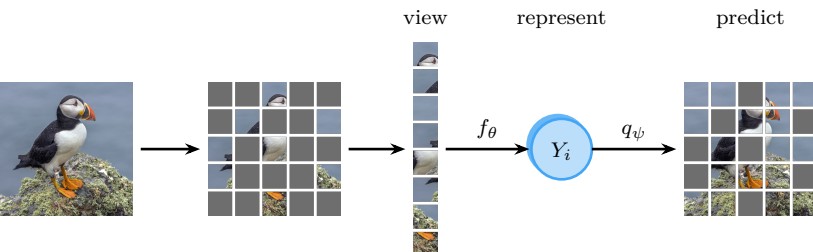

Figure 4: The masked autoencoder predicts masked patches using vision transformers for the encoder and predictor.

Masked autoencoders (He et al., 2022) are similar to denoising autoencoders, where the input images are corrupted and the autoencoder tries to reconstruct the original image. More specifically, the input image is split into smaller non-overlapping patches, from which a random subset is masked out.

Given a batch of images $X$, we consider a single image $x_i$. The image is split up into $m$ patches $X_i = [x_i^{(1)}, \ldots, x_i^{(m)}]$, some of which will be randomly chosen to be masked. We call the set of indices of the masked patches $M_i^{\text{mask}}$ and the set of unmasked indices $M_i^{\text{keep}}$. The encoder $f_\theta$ converts the *unmasked* patches into patch representations $Y_i = [y_i^{(j)} : j \in M_i^{\text{keep}}] = f_\theta([x_i^{(j)} : j \in M_i^{\text{keep}}])$, implemented as a vision transformer (Dosovitskiy et al., 2021). The predictor $q_\psi$ is another vision transformer that predicts the *masked* patches $\hat{X}_i = [\hat{x}_i^{(j)} : j \in M_i^{\text{mask}}] = q_\psi(Y_i, M_i^{\text{mask}})$ from $Y_i$ with a mask token for each index in $M_i^{\text{mask}}$. See Figure 4 for an illustration. The loss is the mean squared error between the pixels of the predicted patches and the

masked patches

$$\mathcal{L}_{\theta,\psi}^{\mathrm{MAE}} = \frac{1}{n} \sum_{i=1}^{n} \sum_{j \in M_i^{\mathrm{mask}}} d_{\mathrm{se}}\left(\hat{x}_i^{(j)}, x_i^{(j)}\right). \tag{17}$$

Without precaution the model could "cheat" by predicting image patches from neighboring pixels, since the information in natural images is usually very spatially redundant. To learn good representations, it is crucial that the masking ratio is very high, e.g., 75%, to encourage the encoder to extract more high-level features.

## 3 Information Maximization Methods

Many self-supervised representation learning methods make use of image transformations. Jigsaw and Rotation Networks, two approaches from the preceding section, apply selected transformations to image examples with the aim to predict the transformation's parametrization. In contrast, the following methods focus on learning representations that are invariant to certain transformations. Such a task typically entails a popular failure mode, called *representation collapse*. It commonly describes trivial solutions, e.g., constant representations, that satisfy the invariance objective, but provide little to no informational value for actual downstream tasks. Instead of learning a rich and diverse feature presentation of the data, the representations *collapse* to a less informative state, often by mapping different or unrelated data points to the same outputs. With that, representation collapse can also be viewed as information collapse in the context of information theory, concentrating a majority of the probability mass of the embedding in a single point, which leads to a decrease of information content. There are several factors that facilitate the occurrence of information collapse, including insufficient regularization and penalization of uninformative representations in the loss function.

To avoid the representation collapse the so-called *information maximization* methods have been developed. They form a class of representation techniques that focus on the information content of the embeddings (Zbontar et al., 2021; Bardes et al., 2021; Ermolov et al., 2021). Due to the employed techniques that aim to maximize information, some of these methods may also be referred to as *feature decorrelation* methods (Tao et al., 2022), as they explicitly decorrelate all elements of embedding vectors. This effectively avoids collapse and results in an indirect maximization of information content. In the following, we present methods that implement this technique using the normalized cross-correlation matrix of embeddings across views (Zbontar et al., 2021), the covariance matrix for single views (Bardes et al., 2021), as well as a whitening operation (Ermolov et al., 2021).

**Transformations.**  The main idea of information maximization methods is that the learned representations should be invariant with respect to certain transformations, i.e., the original image and the transformed images should yield the same representations. We have already encountered two transformations in the previous section, the rotation and jigsaw transformation. There are many more transformations possible: the following transformations have been proven useful for the next methods to be described.

1. Horizontal flipping: Most natural images can be flipped horizontally without changing the semantics, e.g., an image of a car still shows a car after being flipped. Vertical flipping can cause problems when for example the sky is suddenly at the bottom of the image.

2. Blurring: Convolving an image with a Gaussian filter is another way to transform an image.

3. Adding Gaussian noise: Learned representations should also be (to some extent) invariant to the application of noise.

4. Sobel filter: Applying a Sobel filter to an image highlights the edges of an image. These edges usually still contain a lot of relevant information about the image.

5. Cropping and resizing: Scaling the image to a different size should also keep the semantic information.

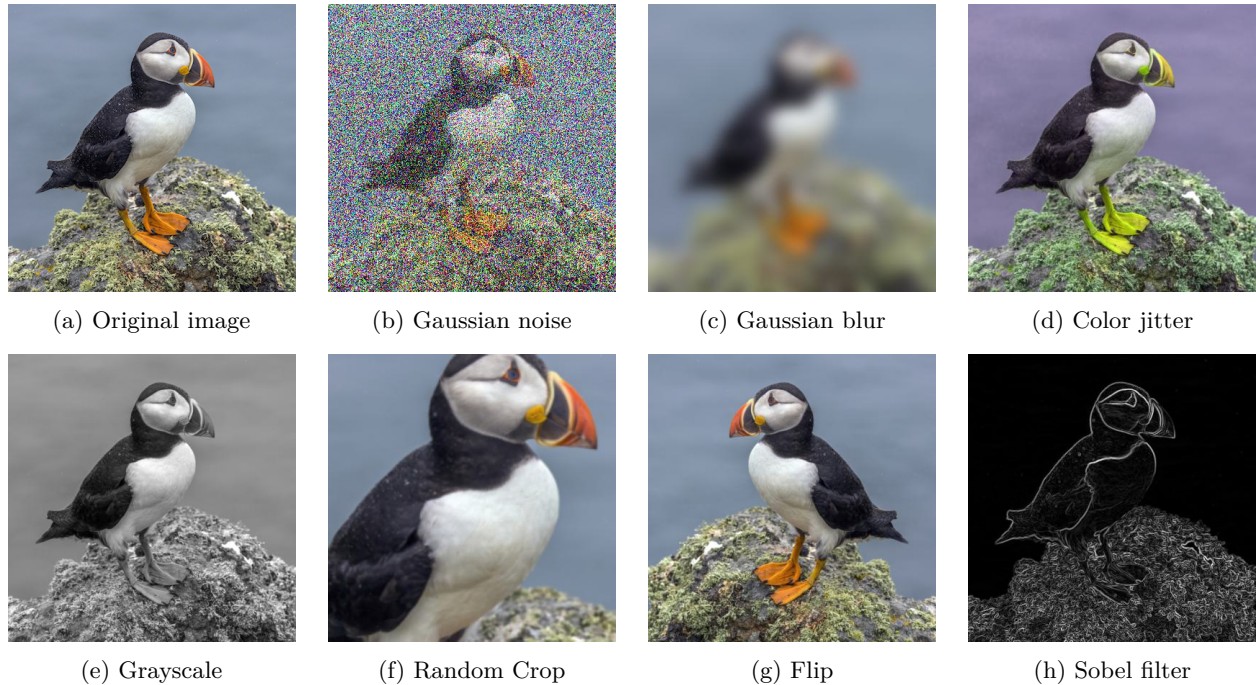

(a) Original image    (b) Gaussian noise    (c) Gaussian blur    (d) Color jitter

(e) Grayscale    (f) Random Crop    (g) Flip    (h) Sobel filter

Figure 5: Example transformations applied to an image of a puffin.

6. Color jittering: Changing the contrast, brightness and hue of an image yields another instance of an image that shows the same content.

7. Grayscaling: Converting color-images to grayscale images is closely related to color jittering.

Note that these image transformations are closely related to dataset augmentation techniques used in supervised learning (Yang et al., 2022; Shorten & Khoshgoftaar, 2019).

## 3.1 Barlow Twins

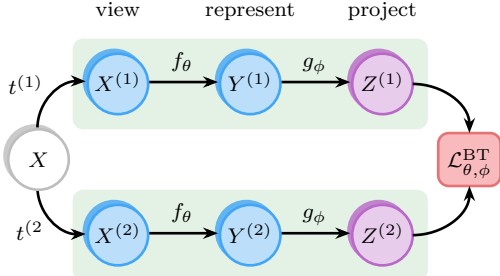

Figure 6: Barlow twins takes two views of the input batch and minimizes the correlation of the projected representations.

The central idea behind this framework is the principle of redundancy reduction. This principle was introduced by the neuroscientist Barlow (1961) and states that the reduction of redundancy is important for the organization of sensory messages in the brain.

To implement this redundancy reduction principle, the Barlow Twins approach takes a batch of images $X$ and creates two views $X^{(1)} = t^{(1)}(X)$ and $X^{(2)} = t^{(2)}(X)$ of these images, where $t^{(1)}, t^{(2)} \sim \mathcal{T}$ are transformations that is randomly sampled from $\mathcal{T}$ for every image of the batch. A Siamese encoder $f_\theta$ computes

representations $Y^{(1)} = f_\theta(X^{(1)})$ and $Y^{(2)} = f_\theta(X^{(2)})$, which are fed into a Siamese projector $g_\phi$ to compute projections $Z^{(1)} = [z_1^{(1)}, \ldots, z_n^{(1)}] = g_\phi(Y^{(1)})$ and $Z^{(2)} = [z_1^{(2)}, \ldots, z_n^{(2)}] = g_\phi(Y^{(2)})$ for both views.

The idea of Barlow Twins is to regularize the cross-correlation matrix between the projections of both views. The cross-correlation matrix is calculated as

$$C = \frac{1}{n} \sum_{i=1}^{n} \left( (z_i^{(1)} - \mu^{(1)})/\sigma^{(1)} \right) \left( (z_i^{(2)} - \mu^{(2)})/\sigma^{(2)} \right)^\top, \tag{18}$$

where $\mu^{(j)}$ and $\sigma^{(j)}$ are the mean and standard deviation over the batch of projections of the $j$-th view, calculated as

$$\mu^{(j)} = \frac{1}{n} \sum_{i=1}^{n} z_i^{(j)}, \tag{19}$$

$$\sigma^{(j)} = \sqrt{\frac{1}{n-1} \sum_{i=1}^{n} (z_i^{(j)} - \mu^{(j)})^2}. \tag{20}$$

The loss function is then defined as

$$\mathcal{L}_{\theta,\phi}^{\text{BT}} = \sum_{k=1}^{d} (1 - C[k,k])^2 + \lambda \sum_{k=1}^{d} \sum_{k' \neq k} C[k,k']^2, \tag{21}$$

where $d$ is the number of dimensions of the projection and $\lambda > 0$ is a hyperparameter. The first term promotes invariance with regard to the applied transformations and the second term decorrelates the learned embeddings, i.e., reduces redundancy. By using this loss, the encoder $f_\theta$ is encouraged to predict embeddings that are decorrelated and thereby non-redundant. The Barlow Twins are trained using the LARS optimizer (You et al., 2017). Note that this loss function is related to the VICReg method, which we are going to introduce next: the first term is called *variance* term and the second *covariance* term.

## 3.2 Variance-Invariance-Covariance Regularization (VICReg)

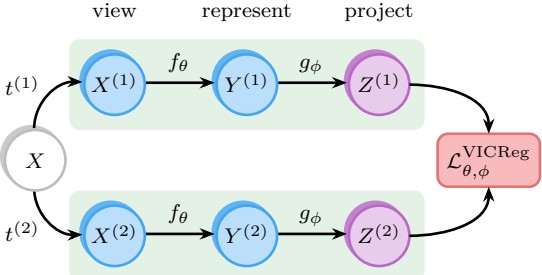

Figure 7: VICReg takes two views of the input batch and minimizes the mean squared error between the projected representations, while regularizing the covariance matrices of representations from individual views to avoid representation collapse.

VICReg was introduced by Bardes et al. (2021) and is a joint-embedding architecture that falls into the category of information maximization methods. Figure 7 gives an overview of the architecture, which is identical to Barlow twins, but uses a different loss function. It aims to maximize agreement between representations of different views of an input, while preventing *informational collapse* using two additional regularization terms. Specifically, VICReg defines regularization terms for variance, invariance and covariance.

Given a batch of images $X$, two views $X^{(1)} = t^{(1)}(X)$ and $X^{(2)} = t^{(2)}(X)$ are defined, where $t^{(1)}, t^{(2)} \sim \mathcal{T}$ are, again, randomly sampled from $\mathcal{T}$ for every image of the batch. A Siamese encoder $f_\theta$ computes representations $Y^{(1)} = f_\theta(X^{(1)})$ and $Y^{(2)} = f_\theta(X^{(2)})$, which are fed into a Siamese projector $g_\phi$ to compute

projections $Z^{(1)} = [z_1^{(1)}, \ldots, z_n^{(1)}] = g_\phi(Y^{(1)})$ and $Z^{(2)} = [z_1^{(2)}, \ldots, z_n^{(2)}] = g_\phi(Y^{(2)})$. Each projection has $d$ dimensions. For each view, the covariance matrix of the projections is computed, which is defined as

$$C^{(j)} = \frac{1}{n-1} \sum_{i=1}^{n} \left( z_i^{(j)} - \mu^{(j)} \right) \left( z_i^{(j)} - \mu^{(j)} \right)^\top, \tag{22}$$

where $\mu^{(j)}$ is the mean over the batch of projections of the $j$-th view, i.e.,

$$\mu^{(j)} = \frac{1}{n} \sum_{i=1}^{n} z_i^{(j)}. \tag{23}$$

The *variance* term aims to keep the standard deviation of each element of the embedding across the batch dimension above a margin $b$. Practically, this prevents embedding vectors to be the same across the batch and thus is one of the two mechanisms that intent to prevent collapse. It can be implemented using a hinge loss

$$\ell_{\text{V}}(Z^{(j)}) = \frac{1}{d} \sum_{k=1}^{d} \max\left( 0, b - \sqrt{C^{(j)}[k,k] + \varepsilon} \right). \tag{24}$$

where $\varepsilon > 0$ is a small hyperparameter for numerical stability. Bardes et al. (2021) used $b = 1$. On that note, the variance term is closely related to the invariance term of Barlow Twins (Zbontar et al., 2021), but applied with a different intention. While Barlow Twins practically maximizes the squared diagonals of the normalized cross-correlation matrix to encourage correlation of embedding elements across views, VICReg maximizes the square root of the diagonals of the covariance matrix of single views in order to prevent collapse. Note, that the maximization in Barlow Twins is restricted by the preceding normalization of the embeddings. As VICReg does not apply a normalization, the margin loss is used to restrict this optimization.

The *covariance* term decorrelates elements of embedding vectors for single views in order to reduce redundancy and prevent collapse. This is achieved by minimizing the squared off-diagonal elements of the covariance matrix $C^{(j)}$ towards 0, i.e.,

$$\ell_{\text{C}}(Z^{(j)}) = \frac{1}{d} \sum_{k=1}^{d} \sum_{k' \neq k} \left( C^{(j)}[k,k'] \right)^2. \tag{25}$$

Note, that this is similar to the redundancy reduction term used in Barlow Twins (Equation 21, right summand), the main difference being again that Barlow Twins applies it across views, but with a similar intention.

Finally, the *invariance* term is used to maximize the agreement between two projections $z_i^{(1)}$ and $z_i^{(2)}$ of the same image, thus inducing invariance to the transformations that were applied to $x_i$. For this, Bardes et al. (2021) apply the mean squared error between the projections

$$\ell_{\text{I}}(Z^{(1)}, Z^{(2)}) = \frac{1}{n} \sum_{i=1}^{n} d_{\text{se}}\left( z_i^{(1)}, z_i^{(2)} \right). \tag{26}$$

Notably, it is the only loss term in VICReg operating across different views.

Overall, the loss of VICReg can be defined as the weighted sum of all three regularizations for the given views

$$\mathcal{L}_{\theta,\phi}^{\text{VICReg}}(X) = \lambda_{\text{V}}[\ell_{\text{V}}(Z^{(1)}) + \ell_{\text{V}}(Z^{(2)})] + \lambda_{\text{C}}[\ell_{\text{C}}(Z^{(1)}) + \ell_{\text{C}}(Z^{(2)})] + \lambda_{\text{I}}\ell_{\text{I}}(Z^{(1)}, Z^{(2)}), \tag{27}$$

where $\lambda_{\text{V}}, \lambda_{\text{I}}, \lambda_{\text{C}} > 0$ are hyperparameters that balance the individual losses.

### 3.3  Whitening (WMSE)

Whitening linearly transforms a set of data points, such that the resulting data points are decorrelated and have unit variance, i.e., the covariance matrix becomes the identity matrix. The method WMSE (Ermolov et al., 2021) applies this idea to the embeddings of images to prevent the representation collapse.

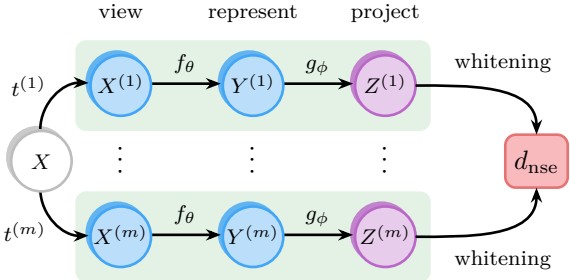

Figure 8: WMSE training: The batch of input images $X$ are randomly transformed and fed into the encoder network $f_\theta$. The representations are then projected using the projection head $g_\phi$. Next, whitening is applied to the projections. The networks are trained by minimizing the normalized MSE between the projections.

Given a batch of images $X$, random transformations $t^{(1)}, \ldots, t^{(m)}$ are applied to obtain $m$ views $X^{(j)}$ for all $j \in \{1, \ldots, m\}$. A Siamese encoder $f_\theta$ maps the views onto representations $Y^{(j)} = f_\theta(X^{(j)})$, which are then fed into a Siamese projector $g_\phi$ to compute projections $Z^{(j)} = [z_1^{(j)}, \ldots, z_n^{(j)}] = g_\phi(Y^{(j)})$. All projections are then concatenated into a single matrix $Z = [z_1^{(1)}, \ldots, z_n^{(1)}, \ldots, z_1^{(m)}, \ldots, z_n^{(m)}]$. This matrix is *whitened* to obtain $\tilde{Z}$ by removing the mean and decorrelating it using the Cholesky decomposition of the inverse covariance matrix, i.e.,

$$\tilde{Z} = [\tilde{z}_1^{(1)}, \ldots, \tilde{z}_n^{(1)}, \ldots, \tilde{z}_1^{(m)}, \ldots, \tilde{z}_n^{(m)}] = W_Z\left(Z - \frac{1}{nm}\sum_{i=1}^{n}\sum_{j=1}^{m} z_i^{(j)}\, \mathbf{1}_{nm}^\top\right), \tag{28}$$

where $\mathbf{1}_{nm}$ is an all-ones vector with $nm$ entries and $W_Z$ is the Cholesky factor of the inverse covariance matrix $P^{ZZ}$ (also known as the precision matrix), i.e., $W_Z W_Z^\top = P^{ZZ}$. Note that the Cholesky decomposition is differentiable which allows to backpropagate through it during training.

To train the models, the normalized squared error between all pairs of the whitened projections is minimized, i.e., the loss function is defined as

$$\mathcal{L}_{\theta,\phi}^{\text{WMSE}} = \frac{1}{n}\sum_{i=1}^{n}\frac{2}{m(m-1)}\sum_{j=1}^{m}\sum_{k=j+1}^{m} d_{\text{nse}}\left(\tilde{z}_i^{(j)}, \tilde{z}_i^{(k)}\right). \tag{29}$$

The constant $2/(m(m-1))$ is the number of comparison per image. The whitening step is essential to prevent the representations from collapsing. The objective maximizes the similarity between all augmented pairs while also preventing representation collapse by enforcing unit covariance on the projections.

**Batch slicing.** One problem of the original method is that the loss has a large variance over consecutive training batches. To counteract this issue, Ermolov et al. (2021) employ so-called batch slicing: the idea of batch slicing is that different views of the same image $z_i^{(1)}, z_i^{(2)}, \ldots, z_i^{(m)}$ should not be in the same batch when the whitening matrix is calculated. For this, $Z$ is partitioned into $m$ parts. The elements of each part are then permuted using the same permutation for each of the $m$ parts. Finally, each of the parts is further subdivided into $d$ subsets which are then used to calculate the whitening matrix for that specific subset. In that way, the loss minimization is dependent on $m \cdot d$ covariance matrices that need to satisfy the identity, leading empirically to lower variance.

### 3.4 Relations to Other Methods

Pretext task methods and information maximization methods both provide training procedures that allow models to learn meaningful and useful features from input data without the need for explicit labels or other additional information. Pretext task methods, like autoencoders, describe mechanisms that already entail relations between data points that qualify for meaningful supervision. In that context, information

maximization methods also explore specific mechanisms that leverage the similarity relation between multiple views of the same example by maximizing mutual information.

## 4 Teacher-Student Methods

Methods based on teacher-student learning are closely related to information maximization methods. Similar to information maximization methods, the student learns to predict the teacher's representations across different image transformations. This allows the student to learn invariant representations that are robust to different transformations of the same image. These methods consist of two branches, where one is considered the student and the other the teacher. To prevent representational collapse as defined in Section 3, the teacher provides stable target representations for the student to predict. To provide stable targets, the teacher is not updated using gradient descent and its parameters are fixed when updating the student. Sometimes a momentum encoder is used between the teacher and student to update the fixed targets. That is, the weights from the student are slowly copied to the teacher to provide more recent targets. The teacher usually has the same architecture as the student, but does not necessarily have the same parameters. The teacher can be a running average of the student's representations, where, e.g., a momentum encoder is used to update the teacher network with the student's weights. For some teacher-student methods the teacher shares the student's weights and an additional predictor network has to predict the teacher's representation.

### 4.1 Bootstrap Your Own Latent (BYOL)

BYOL (Grill et al., 2020) is inspired by the observation that learning representations by predicting fixed representations from a randomly initialized target network avoids representational collapse albeit with sub-par performance. This naturally entails a teacher-student architecture where the teacher (target network) provides stable representations for the student (online network) to learn on.

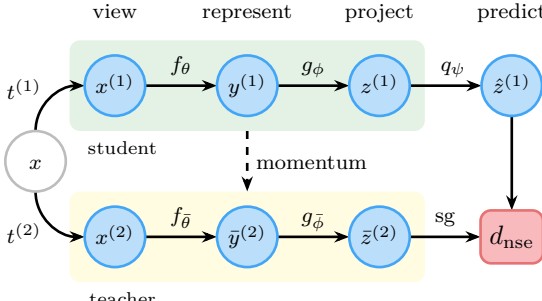

Figure 9: BYOL consists of a student and teacher network. The teacher network is not updated via gradient descent (stop-gradient) and thus provides stable representations for the student network to learn. The teacher network is updated iteratively via an exponential moving average of the student. The student branch has a predictor which is trained to match the fixed representations of the teacher.

BYOL defines two different networks: a student network and a teacher network. The architecture is shown in Figure 9, the student network and teacher network consist of the following parts:

- Student network: encoder $f_\theta$, projector $g_\phi$, predictor $q_\psi$

- Teacher network: encoder $f_{\bar\theta}$, projector $g_{\bar\phi}$

The encoder $f$ and projector $g$ are present in both the student and teacher networks, whereas the predictor $q$ is only part of the student network.

**Learning augmentation-invariant features from the teacher.** Like information-maximization methods, teacher-student methods learn representations by applying different transformations to the images

(see Section 3). Given an image $x_i$, BYOL applies randomly-sampled transformations $t^{(1)}, t^{(2)} \sim \mathcal{T}$ to obtain two different views $x_i^{(1)} = t^{(1)}(x_i)$ and $x_i^{(2)} = t^{(2)}(x_i)$. The student network computes representations $y_i^{(j)} = f_\theta(x_i^{(j)})$, projections $z_i^{(j)} = g_\phi(y_i^{(j)})$, and predictions $\hat{z}_i^{(j)} = q_\psi(z_i^{(j)})$ for both views $j \in \{1, 2\}$. The views are also fed to the teacher network to obtain target projections $\bar{z}_i^{(1)} = g_{\bar\phi}(f_{\bar\theta}(x_i^{(1)}))$ and $\bar{z}_i^{(2)} = g_{\bar\phi}(f_{\bar\theta}(x_i^{(2)}))$.

BYOL minimizes two normalized squared errors: (i) between the prediction of the first view and the target projection of the second view, (ii) between the prediction of the second view and the target projection of the first view. The final loss function is

$$\mathcal{L}_{\theta,\phi,\psi}^{\mathrm{BYOL}} = \frac{1}{n} \sum_{i=1}^{n} \left[ d_{\mathrm{nse}}(\hat{z}_i^{(1)}, \bar{z}_i^{(2)}) + d_{\mathrm{nse}}(\hat{z}_i^{(2)}, \bar{z}_i^{(1)}) \right]. \tag{30}$$

Note that the loss is minimal when the cosine similarity between the vectors is 1. Thus, representations are learned that are similar for two different transformations. In other words, the information content in the learned representations is maximized.

**Teacher-student momentum encoder.** At each training step, the loss is minimized with respect to $\theta$, $\phi$, and $\psi$. That is, only the weights of the student are updated by the gradient of the loss function using the LARS optimizer (You et al., 2017). The weights of the teacher are updated by the exponential moving average (Lillicrap et al., 2019), i.e.,

$$\bar\theta \leftarrow \tau\bar\theta + (1-\tau)\theta, \tag{31}$$

$$\bar\phi \leftarrow \tau\bar\phi + (1-\tau)\phi, \tag{32}$$

where $\tau \in [0, 1]$ controls the rate at which the weights of the teacher network are updated with the weights of the student network.

The authors show that BYOL's success relies on two key components: (i) keeping the predictor $q_\psi$ near optimal at all times by predicting the stable target representations, and (ii) updating the parameters in the direction of $\nabla_{\theta,\phi,\psi}\mathcal{L}^{\mathrm{BYOL}}$ and not in the direction of $\nabla_{\bar\theta,\bar\phi}\mathcal{L}^{\mathrm{BYOL}}$. In other words, the proposed loss and update do not jointly optimize the loss over $\theta, \phi$ and $\bar\theta, \bar\phi$, which would lead to a representation collapse. Chen & He (2021) provide further insight on how this is related to the predictor: regarding (i) it is observed that keeping the learning rate of the predictor fixed instead of decaying it during training improves performance, which supports the fact that the predictor should learn the latest representations. Regarding (ii) Chen & He (2021) use a predictor that maps the projection to the identity. They argue that the gradient of the symmetrized loss with an identity predictor, i.e., between the two projections cancels out the stop-gradient operator. In this case the gradient of the symmetrized loss between the two projections is in the same direction, hence leading to a collapsed representation. Note that this analysis was performed for SimSiam (Chen & He, 2021), which does not use a momentum encoder. It is feasible however that the predictor plays the same role for BYOL.

## 4.2 Self-Distillation With No Labels (DINO)

One of the main contributions of DINO (Caron et al., 2021) is adapting the teacher-student architecture closer to the knowledge distillation framework (Gou et al., 2021), where instead of matching the output embeddings directly, the teacher provides soft labels by applying a softmax operation on its output. The authors show that this facilitates preventing a representation collapse.

DINO defines a student and a teacher network. The student consists of an encoder $f_\theta$ and a projector $g_\phi$ with parameters $\theta$ and $\phi$. The encoder is implemented as a vision transformer (ViT, Dosovitskiy et al., 2021) and the projector as an MLP. The teacher consists of an encoder $f_{\bar\theta}$ and a projector $g_{\bar\phi}$ with the same architecture as the student, but a separate set of parameters $\bar\theta$ and $\bar\phi$.

DINO uses a multi-crop strategy first proposed by Caron et al. (2020) to create a batch of $m$ views $X_i = [x_i^{(1)}, \ldots, x_i^{(m)}]$ of an image $x_i$. Each view is a random crop of $x_i$ followed by more transformations. Most

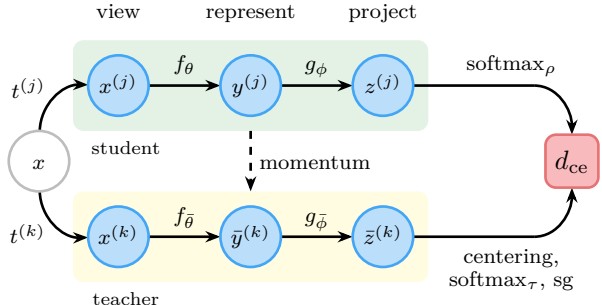

Figure 10: DINO consists of two ViTs, one acting as a student and the other as teacher. The embeddings of the ViT's are transformed into a Softmax distribution which produces soft labels for both the student and the teacher. The knowledge of the student is then iteratively distilled onto the teacher, which provides stable targets for the student.

crops cover a small region of the image, but some crops are of high resolution, which we refer to as *local* and *global* views, respectively. Let $M_i$ be the set of indices of the global views. The idea is that the student has access to all views, while the teacher only has access to the global views, which creates "local-to-global" correspondences (Caron et al., 2021). See Figure 11 for an illustration.

The student computes representations $y_i^{(j)} = f_\theta(x_i^{(j)})$ and projections $z_i^{(j)} = g_\phi(y_i^{(j)})$ for each view. The teacher computes target projections $\bar{z}_i^{(j)} = g_{\bar{\phi}}(f_{\bar{\theta}}(x_i^{(j)}))$ for the global views $j \in M_i$.


Teacher             Student


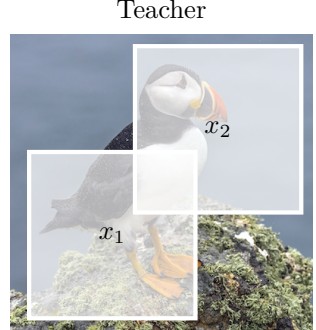 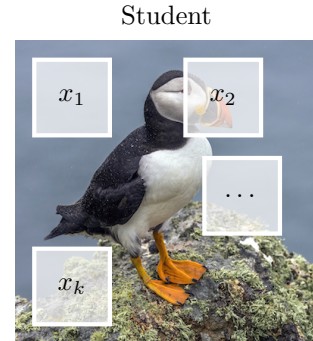

Figure 11: Two sets of augmented crops for the teacher and student as defined by the multi-crop augmentation strategy. The teacher set contains two global views which cover at least 50% of the augmented image. The student set is made up of $k$ patches that cover less than 50% of the other augmented version of the input image.

**Preventing collapse.** Caron et al. (2021) experimentally discover two forms of collapse: Either the computed probability distribution is uniform, or one dimension dominates, regardless of the input. This motivates two countermeasures:

1. To prevent collapse to a uniform distribution, the target distribution of the teacher is *sharpened* by setting the temperature parameter $\tau$ to a small value.

2. To prevent one dimension from dominating, the output of the teacher is centered to make it more uniform. This is accomplished by adding a centering vector $c$ as a bias to the teacher, which is computed with an exponentially moving average

$$c \leftarrow \beta c + (1 - \beta)\bar{z}, \tag{33}$$

where $\beta \in [0, 1]$ is a decay hyperparameter determining to what extent the centering vector is updated and

$$\bar{z} = \frac{1}{n} \sum_{i=1}^{n} \frac{1}{|M_i|} \sum_{j \in M_i} \bar{z}_i^{(j)} \tag{34}$$

is the mean of all projections of the teacher in the current batch.

**Learning invariant features via soft labels.** DINO formulates the task of predicting the target projections as a knowledge distillation task. The projections of the teacher and the student are converted to probability distributions by applying the softmax function over all components. Hereby, the cross-entropy loss can be applied, where the teacher computes soft labels for the student. The total loss function matches every view of the student to every global view of the teacher (except the same global views), i.e.,

$$\mathcal{L}_{\theta,\phi}^{\mathrm{DINO}} = \frac{1}{n} \sum_{i=1}^{n} \sum_{j \in M_i} \sum_{k \neq j} d_{\mathrm{ce}}(\mathrm{softmax}_\tau(\bar{z}_i^{(j)} - c), \mathrm{softmax}_\rho(z_i^{(k)})), \tag{35}$$

where $\tau, \rho > 0$ are hyperparameters controlling the temperature of the distributions (see Section 1.4) for the teacher and the student, respectively. Overall, the parameter updates are very similar to BYOL's, since the student network is updated by minimizing the loss $\mathcal{L}_{\theta,\phi}^{\mathrm{DINO}}$ using the AdamW optimizer, and the teacher network is updated by an exponential moving average of the student, i.e.,

$$\bar{\theta} \leftarrow \alpha\bar{\theta} + (1 - \alpha)\theta, \tag{36}$$

$$\bar{\phi} \leftarrow \alpha\bar{\phi} + (1 - \alpha)\phi, \tag{37}$$

where $\alpha \in [0, 1]$ controls the rate at which the weights of the teacher network are updated with the weights of the student network.

The authors identify interesting properties when training ViTs in a self-supervised manner. ViTs trained via self-supervision are able to detect object boundaries within a scene layout which is information that can be extracted within the attention layers. Furthermore, the learned attention maps learn segmentation masks, i.e., the objects are separated from the background in the attention masks. These attention masks allow DINO to perform well on downstream tasks simply by using a k-nearest-neighbor classifier on its representations.

### 4.3 Efficient Self-Supervised Vision Transformers (EsVit)

Li et al. (2021) keep the same teacher-student architecture as Caron et al. (2021), but replace the ViTs in Caron et al. (2021) with multi-stage transformers. As an optimization to the transformer architecture, a multi-stage transformer subsequently merges image patches together across every layer to reduce the number of image patches that have to be processed. The authors show that the merging process destroys important local to global correspondences which are learned in common transformers. Therefore, an additional region matching loss is proposed that mitigates the lost semantic correspondences during the merging process in multi-stage transformer architectures.

**Multi-stage vision transformers.** Vaswani et al. (2021) reduce computational complexity of a standard transformer by reducing the number of patches that go through the transformer at each layer. For this, a special image patch merging module merges the patches at each layer and attention is calculated between them via sparse self-attention modules. This process is repeated multiple times. An illustration of the procedure is provided in Figure 12. Overall, the number of tokens, i.e., feature maps that have to be evaluated by one self-attention module is reduced through each subsequent layer, while allowing for more diverse feature learning due to the different self-attention heads processing different merged patches, allowing for hierarchical correspondences to be learned.

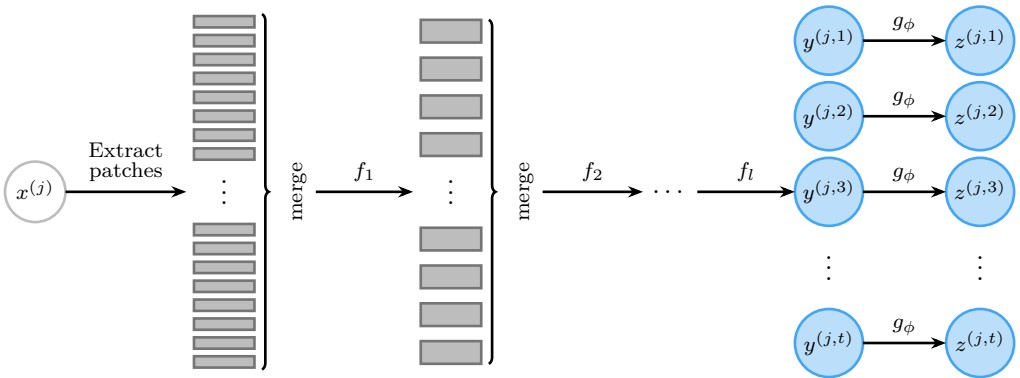

Figure 12: Multi-stage transformer: An image is decomposed into image patches, which are merged and applied self-attention to subsequently. This results in a smaller number of patches being processed simultaneously, while also learning hierarchical embeddings. At the end, either average pooling is performed over the outputs or the output sequence is used as it is.

**Extending the loss function for region level feature matching.** Li et al. (2021) propose to extend DINO's loss for multi-stage transformers in order to learn the local-to-global correspondences lost during the merging process. As EsVit is an extension of DINO, the teacher network and student network are defined as in Section 4.2. Li et al. (2021) propose a loss function that consists of a view-level loss $\ell_{\text{view}}$ and a region-level loss $\ell_{\text{region}}$. The view-level loss is the same loss used to train DINO, i.e.,

$$\ell_{\text{view}} = \frac{1}{n} \sum_{i=1}^{n} \sum_{j \in M_i} \sum_{k \neq j} d_{\text{ce}}(\text{softmax}_\tau(\bar{z}_i^{(j)} - c_{\text{view}}), \text{softmax}_\rho(z_i^{(k)})). \tag{38}$$

where $c_{\text{view}}$ is the centering vector for the view-level loss.

The region-level loss of EsVit is computed from the encoder outputs for each image patch $Y_i = [y_i^{(1,1)}, \dots, y_i^{(m,T)}]$ directly (see Figure 12), where $T$ is the sequence length, i.e., the number of patches for a given view $j$. Then the region-level loss is defined as

$$\ell_{\text{region}} = \frac{1}{n} \sum_{i=1}^{n} \sum_{j \in M_i} \sum_{k \neq j} \sum_{t=1}^{T} d_{\text{ce}}(\text{softmax}_\tau(\bar{z}_i^{(j,s^*)} - c_{\text{region}}), \text{softmax}_\rho(z_i^{(k,t)})). \tag{39}$$

where $s^* = \text{argmax}_s s_{\cos}(\bar{z}^{(j,s)}, z^{(k,t)})$, $T$ is the number of image patches, and $c_{\text{region}}$ is the centering vector for the region-level loss. The idea is to match every image patch projection of the student $z^{(k,t)}$ to the best image patch projection of the teacher $\bar{z}^{(j,s^*)}$. That is, for every projection as defined by the multi-crop strategy in Figure 11, the region-level loss matches the most concurring image patches of the student and teacher. The final EsVit loss combines the view-level and region-level loss, i.e.,

$$\mathcal{L}_{\theta,\phi}^{\text{EsVit}} = \ell_{\text{view}} + \ell_{\text{region}} \tag{40}$$

The authors show that when only training with the view-level loss on a multi-stage transformer, the model fails to capture meaningful correspondences, such as the background being matched for two augmentations of the same image. Adding the region-level loss mitigates the issue of lost region-level correspondences in multi-stage transformer architectures and recovers some of the correspondences lent inherently by monolithic transformer architectures.

## 4.4 Simple Siamese Representation Learning (SimSiam)

SimSiam was introduced by Chen & He (2021) and uses a similar architecture and loss function as BYOL. However, teacher and student share the same parameters and hence a momentum encoder is not used as in previously presented teacher-student methods.

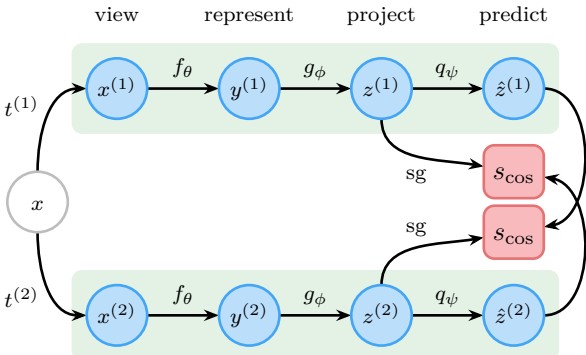

Figure 13: SimSiam maximizes the cosine similarity between the projected representations. Both networks use the same parameters. The stop gradient operator interrupts the backward-flow of the gradients and thereby preventing the representational collapse.

Given a batch of images $X$, for each image $x_i$ two views $x_i^{(1)} = t^{(1)}(x_i)$, $x_i^{(2)} = t^{(2)}(x_i)$ are created using random transformations $t^{(1)}, t^{(2)} \sim \mathcal{T}$ that are sampled for each image of the batch. For each of these views, a Siamese encoder $f_\theta$ computes a representation $y_i^{(j)} = f(x_i^{(j)})$ and a Siamese projector $g_\phi$ computes a projection $z_i^{(j)} = g_\phi(y_i^{(j)})$. Finally, the projection is fed through a predictor $q_\psi$ to obtain a prediction $\hat{z}_i^{(j)} = q_\psi(z_i^{(j)})$.

The goal of the predictor is to predict the projection of the other view. Therefore, the loss computes the negative cosine similarity between the prediction of the first view and the projection of the second view, and vice versa, i.e.,

$$\mathcal{L}_{\theta,\phi,\psi}^{\text{SimSiam}} = -\frac{1}{n} \sum_{i=1}^{n} \frac{1}{2} \left[ s_{\cos}(\hat{z}_i^{(1)}, \text{sg}(z_i^{(2)})) + s_{\cos}(\hat{z}_i^{(2)}, \text{sg}(z_i^{(1)})) \right], \tag{41}$$

where $\text{sg}(\cdot)$ is the stop gradient operator that prevents gradients from being backpropagated through this branch of the computational graph.

The encoder $f_\theta$ is implemented as a ResNet (He et al., 2016). The projector $g_\phi$ and the predictor $q_\psi$ are MLPs. The authors show empirically that a predictor is crucial to avoid collapse. Chen & He (2021) argue that the gradient of the symmetrized loss with a predictor that is the identity is in the same direction as the gradient of the symmetrized loss between the two projections, such that the stop-gradient operation is cancelled out, thus leading to representation collapse. Using a random predictor does not work either and Chen & He (2021) argue that the predictor should always learn the latest representations. This argument is similar to Grill et al. (2020) in Section 4.1, where the predictor should be kept near optimal at all times.

Another crucial ingredient to their method is batch normalization (Ioffe & Szegedy, 2015), which is used for both $f_\theta$ and $g_\phi$. Furthermore, the authors experiment with the training objective by replacing it with the cross-entropy loss. Their experiments show that this also works, however the performance is worse. The key advantage of SimSiam is that training does not require large batch sizes allowing the use of SGD instead of LARS.

## 4.5 Relations to Other Methods

**Information maximization methods.** Recent work has shown that BYOL and SimSiam perform feature decorrelation like information maximization methods such as Barlow Twins and VICReg. To show this, Liu et al. (2022) first replace the prediction MLP $q(z)$ in BYOL and SimSiam by a linear mapping $W_q z$. They then derive an equation for the weight dynamics of a mini-batch,

$$W_q(t)^T W_q(t) = Z(t)Z(t)^T \tag{42}$$

where $W$ is a weight matrix, $Z$ is the mini-batch of projections for the predictor and $t$ defines a time-step. Note that $Z(t)Z(t)^T = \frac{1}{n}\sum_{i=1}^{n} z_i z_i^T$ is the uncentered covariance of the mini-batch $Z$. Liu et al. (2022) show empirically that during training $W_q(t)^T W_q(t)$ becomes diagonal, hence the covariance matrix of the projections $Z(t)Z(t)^T$ becomes diagonal. This means that as training progresses with BYOL and SimSiam, the projections of the examples in the mini-batch become increasingly uncorrelated. Incidentally, this is the same as in the presented feature decorrelation methods. VICReg's main objective is identical, to approximate an identity covariance matrix between samples in the mini-batch, while Barlow Twins similarly approximates an identity cross-correlation matrix between the mini-batches of each image transformation branch.

**Contrastive methods.** Recent work has shown that teacher-student methods seem to perform implicit contrastive learning. We expand on this in Section 5.7.

## 5 Contrastive Methods

Contrastive methods prevent representation collapse by *decreasing* the similarity between representations of unrelated data points. Given a data point $x^*$ called the *anchor*, one defines mechanisms to generate *positive* samples and *negative* samples for the anchor. The positives should retain the relevant information of the anchor, while the negatives should contain information different from the anchor. For vision tasks, the positives could, e.g., be random transformations of the same image, while the negatives are (transformations of) other images. The goal of contrastive methods is to move representations of positives closer to the representation of the anchor while moving representations of negatives away from the anchor.

More formally, given an anchor $x^*$ we define the conditional distributions of positives $p_{\text{pos}}(x^+|x^*)$ and negatives $p_{\text{neg}}(x^-|x^*)$. These distributions are induced by the mechanisms that generate the positives and negatives and are not explicitly known. Let $y^* = f_\theta(x^*)$, $y^+ = f_\theta(x^+)$, and $y^- = f_\theta(x^-)$ be the corresponding representations, calculated with an encoder $f_\theta$ parameterized by $\theta$. The task of contrastive methods is to maximize the likelihood of positive representations $p_{\text{pos}}(y^+|y^*)$ while minimizing the likelihood of negative representations $p_{\text{neg}}(y^-|y^*)$. Note that we continue to use representations $y$ in this introduction, but the same methods can be applied to projections $z$ equivalently.

**Noise contrastive estimation (NCE).** The idea of noise contrastive estimation (Gutmann & Hyvärinen, 2010) is to formulate the task of contrastive representation learning as a supervised classification problem. One assumption of NCE is that the negatives are independent from the anchor, i.e., $p_{\text{neg}}(x^-|x^*) = p_{\text{neg}}(x^-)$. In this context, the negatives are often called *noise*. There are two widely used approaches, the original NCE and InfoNCE (van den Oord et al., 2018). Roughly speaking, NCE performs binary classification to decide whether an individual sample is a positive or negative, whereas InfoNCE performs multiclass classification on a set of samples to decide which one is the positive. In the following we will explain InfoNCE in more detail.

**InfoNCE.** For each anchor $x^*$, InfoNCE generates one positive sample from $p_{\text{pos}}(x^+|x^*)$ and $n-1$ negative samples from $p_{\text{neg}}(x^-)$. Let $X = [x_1, \dots, x_n]$ be the set of those samples, where $x_c$ is the positive with index $c \in \{1, \dots, n\}$. In the context of representation learning we further compute representations using an encoder $f_\theta$ and obtain the set $Y = [y_1, \dots, y_n]$.

InfoNCE now defines a supervised classification task, where the input is $(y^*, Y)$ and the class label is the index of the positive $c$. A classifier $p_\psi(c|Y, y^*)$ with parameters $\psi$ is trained to match the true data distribution of the labels $p_{\text{data}}(c|Y, y^*)$. A common supervised learning objective is to minimize the cross-entropy between the data distribution and the model distribution, i.e.,

$$\min_{\psi,\theta} \mathbb{E}_{Y,y^*} \left[ H(p_{\text{data}}(c|Y, y^*), p_\psi(c|Y, y^*)) \right] \tag{43}$$

$$= \min_{\psi,\theta} \mathbb{E}_{Y,y^*} \left[ \mathbb{E}_{c|Y,y^*} \left[ -\log p_\psi(c|Y, y^*) \right] \right]. \tag{44}$$

Note that this is an anticausal prediction problem, where the underlying cause (label) is predicted from its effect (input) (Schölkopf et al., 2012). In InfoNCE we know the underlying mechanism (since we generate the labels artificially), so we can derive the optimal classifier using Bayes' theorem.

First, we write down the data distribution of a set $Y$ given a label and an anchor, i.e.,

$$p_{\text{data}}(Y|c,y^*) = \prod_{i=1}^{n} p_{\text{data}}(y_i|c,y^*) = \prod_{i=1}^{n} \begin{cases} p_{\text{pos}}(y_i|y^*), & \text{if } i = c, \\ p_{\text{neg}}(y_i), & \text{if } i \neq c, \end{cases} \tag{45}$$

$$= p_{\text{pos}}(y_c|y^*) \prod_{i \neq c} p_{\text{neg}}(y_i) = \frac{p_{\text{pos}}(y_c|y^*)}{p_{\text{neg}}(y_c)} \prod_{i=1}^{n} p_{\text{neg}}(y_i), \tag{46}$$

where we assume conditional independence among the samples in $Y$. InfoNCE further assumes that the labels are sampled uniformly, i.e., $p_{\text{data}}(c) = \frac{1}{n}$. Now we apply Bayes' theorem:

$$p_{\text{data}}(c|Y,y^*) = \frac{p_{\text{data}}(Y|c,y^*)\,p_{\text{data}}(c)}{\sum_{c'=1}^{n} p_{\text{data}}(Y|c',y^*)\,p_{\text{data}}(c')} \tag{47}$$

$$= \frac{\frac{p_{\text{pos}}(y_c|y^*)}{p_{\text{neg}}(y_c)} \prod_{i=1}^{n} p_{\text{neg}}(y_i) \frac{1}{n}}{\sum_{c'=1}^{n} \frac{p_{\text{pos}}(y_{c'}|y^*)}{p_{\text{neg}}(y_{c'})} \prod_{i=1}^{n} p_{\text{neg}}(y_i) \frac{1}{n}} \tag{48}$$

$$= \frac{\frac{p_{\text{pos}}(y_c|y^*)}{p_{\text{neg}}(y_c)}}{\sum_{c'=1}^{n} \frac{p_{\text{pos}}(y_{c'}|y^*)}{p_{\text{neg}}(y_{c'})}}. \tag{49}$$

An optimal classifier with zero cross-entropy would match this distribution. We can see that the optimal probability of a class is the density ratio $\frac{p_{\text{pos}}(y_c|y^*)}{p_{\text{neg}}(y_c)}$, normalized across all classes. It describes the likelihood of $y_c$ being a positive sample for $y^*$ versus being a negative sample. This motivates the choice of the classifier of InfoNCE, which is defined similar to Equation 49:

$$p_{\psi}(c|Y,y^*) = \frac{s_{\psi}(y^*,y_c)}{\sum_{c'=1}^{n} s_{\psi}(y^*,y_{c'})}, \tag{50}$$

where $s_{\psi}(y^*,y)$ is a predictor that computes a real-valued positive score. Minimizing the cross-entropy from Equation 43 brings the model distribution $p_{\psi}(c|Y,y^*)$ closer to the data distribution $p_{\text{data}}(c|Y,y^*)$, which ensures that $s_{\psi}$ approaches the density ratio of the data, i.e., $s_{\psi}(y^*,y) \approx \frac{p_{\text{pos}}(y|y^*)}{p_{\text{neg}}(y)}$ (in fact, it only needs to be proportional to the density ratio). The density ratio is high for positive samples and close to zero for negative samples, which means that $s_{\psi}(y^*,y)$ learns some similarity measure between the representations. Since $\psi$ and $\theta$ (i.e., predictor and encoder) are optimized jointly, the encoder is encouraged to learn similar embeddings for an anchor and its positive, and to learn dissimilar embeddings for an anchor and its negative samples (as long as the predictor is not too expressive). In other words, the encoder is encouraged to extract information that is "unique" to the anchor and the positive sample.

By combining the negative logarithm and the classifier (Equations 44 and 50) the general InfoNCE loss for $(y^*,Y,c)$ is defined as

$$\text{InfoNCE}_{s_{\psi}}(y^*,Y,c) = -\log\left(\frac{s_{\psi}(y^*,y_c)}{\sum_{c'=1}^{n} s_{\psi}(y^*,y_{c'})}\right). \tag{51}$$

For the sake of notation in the following sections, we slightly adjust this definition. All considered methods compute the exponential of some score to obtain positive values for $s_{\psi}$, which is why we include it directly in the loss function. In this case the InfoNCE loss computes the commonly used softmax cross-entropy. Instead of specifying the class label we denote the positive by $y^+$ and the set of negatives by $\bar{Y}$. Thus, our final definition of the InfoNCE loss for a score function $s_{\psi}(y^*,y)$ is

$$\text{InfoNCE}_{s_{\psi}}(y^*,y^+,\bar{Y}) = -\log\left(\frac{\exp(s_{\psi}(y^*,y^+))}{\exp(s_{\psi}(y^*,y^+)) + \sum_{\bar{y} \in \bar{Y}} \exp(s_{\psi}(y^*,\bar{y}))}\right). \tag{52}$$

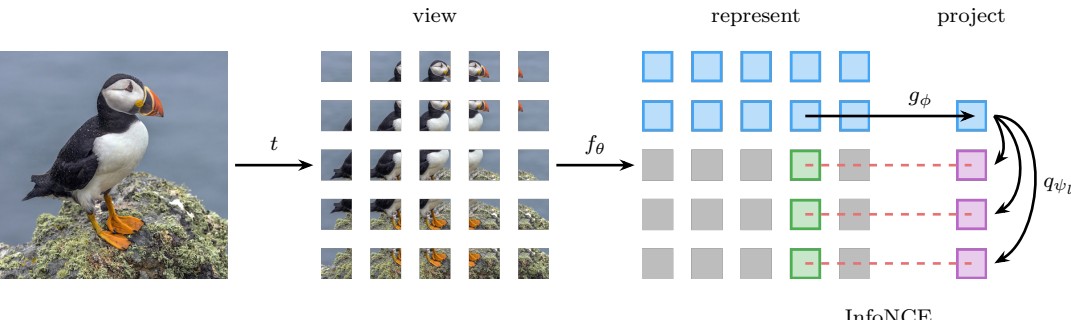

Figure 14: In CPC training an image is transformed and split into overlapping patch encodings, where the top rows (blue) act as the anchor. The positive samples (purple) lie in a column below the exact anchor position and every other patch of the whole dataset is a negative. InfoNCE is applied to distinguish between both distributions.

### 5.1 Contrastive Predictive Coding (CPC)

CPC (van den Oord et al., 2018) is an influential self-supervised representation learning technique which is applicable to a wide variety of input modalities such as text, speech and images. It is based on the theory of predictive coding, which originated in the neuroscience literature by observing the learning behavior of biological neural circuits (Huang & Rao, 2011; Bastos et al., 2012). In short, a model tries to predict future outcomes given the past or *context*. Thus, the learned representation of the context should incorporate all information necessary for prediction while removing unimportant noise. This predictive coding task is solved by formulating it as a contrastive learning problem. CPC operates on sequential data, which is a natural choice for audio data, but can also be applied to vision tasks by splitting images into sequences of patches.

Given a batch of images $X$, we consider a single image $x_i$. The image is split into $m$ patches $[x_i^{(1)}, \ldots, x_i^{(m)}]$. Note that the patches are overlapping and that an additional image augmentation is applied to each patch. A Siamese encoder $f_\theta$ converts each patch into a representation $y_i^{(j)} = f_\theta(x_i^{(j)})$. The *context* of a patch are the patches in the same row and all patches in the rows above. Let $C_i^{(j)} \subset \{1, \ldots, m\}$ be the set of indices of those patches. A projector $g_\phi$ computes a context representation $z_i^{(j)} = g_\phi([y_i^{(k)} : k \in C_i^{(j)}])$ for each patch. The network $g_\phi$ is implemented by a masked CNN (PixelCNN, van den Oord et al., 2016) that restricts its receptive field to the patches in the context.

Given a context representation $z_i^{(j)}$, the predictive coding task of CPC is to predict representations of future patches, which are the patches in the column below the patch. Let $F_i^{(j)} = [k_{i,1}^{(j)}, \ldots, k_{i,K}^{(j)}] \subset \{1, \ldots, m\}$ be the set of indices of those patches. The prediction task is solved by minimizing an InfoNCE loss for each future representation. For the $l$-th future representation with patch index $k = k_{i,l}^{(j)}$, a separate predictor $q_{\psi_l}$ computes the anchor $\hat{y}_i^{(k)} = q_{\psi_l}(z_i^{(j)})$ from the context representation, and the positive is the future representation $y_i^{(k)}$. The negatives can be any unrelated representations, e.g., the representations of all patches outside the context and the future, as well as all representations from other images in $X$. We denote the set of negatives by $\bar{Y}_i^{(j)}$. The loss function accumulates the InfoNCE losses over all contexts and futures across the batch, i.e.,

$$\mathcal{L}_{\theta,\phi,\psi}^{\text{CPC}} = \frac{1}{n} \sum_{i=1}^{n} \frac{1}{m} \sum_{j=1}^{m} \sum_{k \in F_i^{(j)}} \text{InfoNCE}_s(\hat{y}_i^{(k)}, y_i^{(k)}, \bar{Y}_i^{(j)}), \tag{53}$$

where the similarities are calculated using the dot product $s(\hat{y}, y) = \hat{y}^\top y$, and the parameters of all predictors are combined in $\psi = [\psi_1, \ldots, \psi_K]$. See Figure 14 for an illustration of the method.

**CPC v2.** The second version of CPC (Henaff, 2020) targets the problem of sample efficiency. So far, contrastive methods require a large amount of data, especially the need to find hard negatives, to perform

well on common benchmarks such as ImageNet (Bardes et al., 2021). This work focuses on the improvement of training routines rather than modifying the general idea behind CPC. The most notable changes are improved image augmentations, larger network sizes, using layer normalization (Ba et al., 2016) instead of batch normalization (Ioffe & Szegedy, 2015), which creates an unintentional co-dependency between the patches, and extending the prediction task to all four directions, instead of predicting bottom from top patches only.

## 5.2 Contrastive Multiview Coding (CMC)

CMC (Tian et al., 2020) considers multiple views of the same scene and tries to maximize the mutual information between those views. For each view, a view-specific encoder extracts view-invariant features. A contrastive learning objective forces the encoders to be as informative as possible about the other views. In general, the views could be different sensory inputs of the same scene (e.g., color, depth, surface normals). For vision tasks that only have access to RGB images, the different views could be individual color channels. In this paper, the authors consider the L and ab channels of an image after converting it to the Lab color space. Note that the views can be interpreted as image augmentations, however, each view uses the *same* image augmentation for all images, which is in contrast to other methods.

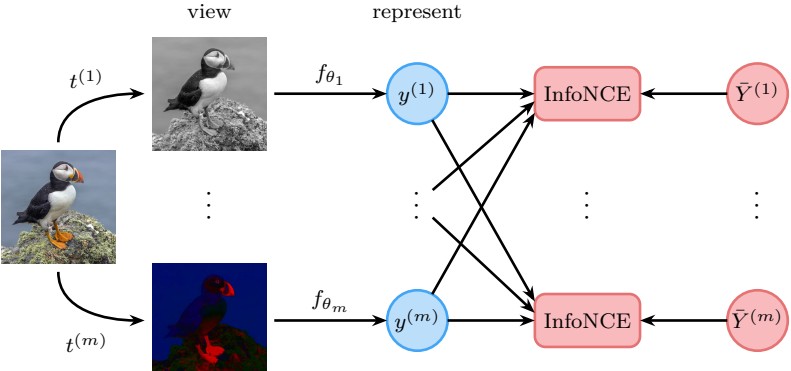

Figure 15: CMC assumes there are different interfaces to the world, that collect information about a shared source. The goal is to find a consensus about the perceived information coming from the different modalities. This is done by extracting the view-invariant features in order to learn a shared representation.

To apply CMC to images, we define $m$ fixed image transformations $[t^{(1)}, \ldots, t^{(m)}]$. Let $X$ be a batch of images that gets transformed into $X^{(j)} = t^{(j)}(X)$ for each view $j \in \{1, \ldots, m\}$. The encoders $f_{\theta_1}, \ldots, f_{\theta_m}$ compute view-specific representations $y_i^{(j)} = f_{\theta_j}(x_i^{(j)})$ for each $j \in \{1, \ldots, m\}$. The representation used for downstream tasks can either be the representation of a specific view or the concatenation of representations across multiple or all views.

The idea of CMC is to apply the InfoNCE loss to pairs of views. Specifically, the anchor is the representation $y_i^{(j)}$ of the $j$-th view, the positive is the representation $y_i^{(k)}$ of the same image but from the $k$-th view, where $k \neq j$, and the negatives are representations from other images but also from the $k$-th view. We denote the set of those negatives images by $\bar{Y}_i^{(k)}$ which are obtained using a memory bank (Wu et al., 2018). With memory banks, large batches of negative samples can be obtained efficiently, at the cost of slightly outdated representations.

The loss for a single image $x_i$ accumulates the InfoNCE losses of all ordered pairs of views, so the total loss function across the batch is given as

$$\mathcal{L}_\theta^{\text{CMC}} = \frac{1}{n} \sum_{i=1}^{n} \sum_{j=1}^{m} \sum_{k=1; \, k \neq j}^{m} \text{InfoNCE}_{s_\tau}(y_i^{(j)}, y_i^{(k)}, \bar{Y}_i^{(k)}), \tag{54}$$

where the similarities are calculated as $s_\tau(y, y') = s_{\cos}(y, y')/\tau$, i.e., as cosine similarity divided by a temperature hyperparameter $\tau > 0$, and where $\theta = [\theta_1, \ldots, \theta_m]$ combines the parameters of all encoders. See Figure 15 for an illustration of the method.

## 5.3 Simple Contrastive Learning of Representations (SimCLR)

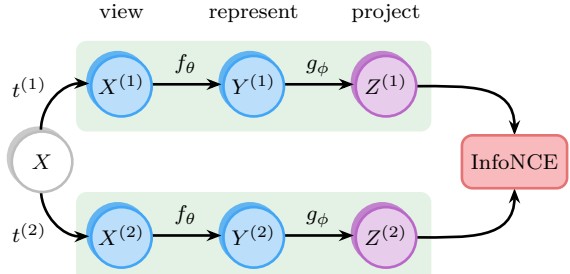

Figure 16: SimCLR defines the views in the batch that were constructed from different images as negative examples.

The architecture used for SimCLR (Chen et al., 2020a) is similar to previous methods like VICReg or Barlow Twins. Given a batch of images $X$, two views $X^{(1)} = t^{(1)}(X)$ and $X^{(2)} = t^{(2)}(X)$ are created using random transformations $t^{(1)}, t^{(2)} \sim \mathcal{T}$. A Siamese encoder $f_\theta$ calculates representations $Y^{(1)} = f_\theta(X^{(1)})$ and $Y^{(2)} = f_\theta(X^{(2)})$ which are then fed into a Siamese projector $g_\phi$ to obtain projections $Z^{(1)} = [z_1^{(1)}, \ldots, z_n^{(1)}] = g_\phi(Y^{(1)})$ and $Z^{(2)} = [z_1^{(2)}, \ldots, z_n^{(2)}] = g_\phi(Y^{(2)})$. Figures 16 gives an overview over this process.

SimCLR uses a contrastive loss to maximize the similarity between the two projections of the same image while minimizing the similarity to projections of other images. Specifically, for an image $x_i$ two InfoNCE losses are applied. The first one uses the anchor $z_i^{(1)}$, the positive $z_i^{(2)}$, and the negatives $\bar{Z}_i = [z_1^{(1)}, z_1^{(2)}, \ldots, z_n^{(1)}, z_n^{(2)}] \setminus \{z_i^{(1)}, z_i^{(2)}\}$, which are all projections from other images in the batch. The second InfoNCE loss swaps the roles of anchor and positive but uses the same set of negatives. Therefore, the loss function is defined as

$$\mathcal{L}_{\theta,\phi}^{\text{SimCLR}} = \frac{1}{n} \sum_{i=1}^{n} \frac{1}{2} \Big[ \text{InfoNCE}_{s_\tau}(z_i^{(1)}, z_i^{(2)}, \bar{Z}_i) + \text{InfoNCE}_{s_\tau}(z_i^{(2)}, z_i^{(1)}, \bar{Z}_i) \Big], \tag{55}$$

where the similarities are calculated as $s_\tau(z, z') = s_{\cos}(z, z')/\tau$, i.e., the cosine similarity divided by a temperature hyperparameter $\tau > 0$.

The transformations consist of a random cropping followed by a resize back to the original size, a random color distortion, and a random Gaussian blur. A ResNet is used as encoder $f_\theta$ and the projector $g_\phi$ is implemented as an MLP with one hidden layer. To train SimCLR large batch sizes are used in combination with the LARS optimizer (You et al., 2017). The authors note that their method does not need memory banks (Wu et al., 2018) as it is the case for other contrastive methods and is thus easier to implement.

## 5.4 Momentum Contrast (MoCo)

Momentum Contrast (He et al., 2020) is a contrastive learning approach that uses a momentum encoder with an encoding queue to bridge the gap between contrastive *end-to-end* and *memory bank* methods (Wu et al., 2018). In essence, it allows the optimization of a contrastive objective with significantly reduced computational costs, both in terms of time and GPU memory (Chen et al., 2020b). Figure 17 gives an overview of the architecture.

Similar to teacher-student methods (Section 4), MoCo defines a student network consisting of an encoder $f_\theta$ and a projector $g_\phi$ with parameters $\theta$ and $\phi$, and a teacher network consisting of an encoder $f_{\bar{\theta}}$ and a projector $g_{\bar{\theta}}$ with parameters $\bar{\theta}$ and $\bar{\phi}$. Given an image $x_i$, two views $x_i^* = t^*(x_i)$ and $x_i^+ = t^+(x_i)$ are created using

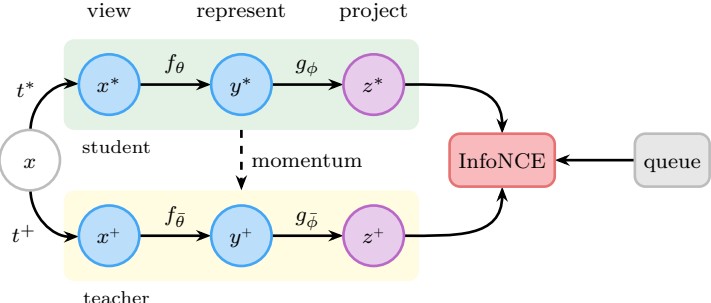

Figure 17: MoCo computes projections of different views with a student and a teacher network, and minimizes the contrastive InfoNCE loss. The projections $z^*$ and $z^+$ are computed on-the-fly, whereas negative projections are cached in a queue of the most recent versions of $z^+$ from previous iterations, significantly improving computational efficiency.

random transformations $t^*, t^+ \sim \mathcal{T}$. The student computes the representation $y_i^* = f_\theta(x_i^*)$ and projection $z_i^* = g_\phi(y_i^*)$, while the teacher computes the representation $y_i^+ = f_{\bar\theta}(x_i^+)$ and projection $z_i^+ = g_{\bar\phi}(y_i^+)$.

MoCo minimizes the InfoNCE loss to learn projections that are similar for two views of the same image and dissimilar to projections of views of other images. In our notation, the student computes the anchor $z_i^*$, the teacher computes the positive $z_i^+$, and the selection of the negatives $\bar{Z}_i$ is described below. The loss of MoCo is then defined as

$$\mathcal{L}_{\theta,\phi}^{\mathrm{MoCo}} = \frac{1}{n} \sum_{i=1}^{n} \mathrm{InfoNCE}_{s_\tau}(z_i^*, z_i^+, \bar{Z}_i), \tag{56}$$

where the similarities are calculated using the dot product $s_\tau(z^*, z) = z^\top z^* / \tau$ divided by a temperature hyperparameter $\tau > 0$. The teacher is updated by an exponential moving average of the student, i.e.,

$$\bar\theta \leftarrow \alpha\bar\theta + (1 - \alpha)\theta, \tag{57}$$

$$\bar\phi \leftarrow \alpha\bar\phi + (1 - \alpha)\phi \tag{58}$$

where $\alpha \in [0, 1]$ controls the rate at which the weights of the teacher network are updated with the weights of the student network.

In an *end-to-end* setting, the negatives are computed on-the-fly in one batch (see SimCLR, Section 5.3), resulting in relatively large resource consumption. In contrast, memory banks (Wu et al., 2018) describe the concept of saving projections for all items in the dataset, drastically reducing resource consumption but introducing potential negative effects from inconsistent or outdated projections. MoCo aims to combine the benefits of both end-to-end training and memory banks. Similar to memory banks, MoCo only computes projections of the positives and saves them for reuse in later iterations. Instead of saving projections for all images in the dataset, MoCo uses a queue to cache only the last $K$ computed projections, thus avoiding outdated projections. Since older projections are removed from the queue, saved projections no longer require momentum updates. The teacher provides the projections that are to be cached, while the student is updated via backpropagation using the contrastive loss.

**MoCo v2.** The second version of MoCo (Chen et al., 2020b) introduces several smaller changes to further improve downstream performance and outperform SimCLR. The most notable changes include the replacement of the linear projection layer of MoCo with an MLP, as well as the application of a cosine learning rate scheduler (Loshchilov & Hutter, 2017) and additional augmentations. The new 2-layer MLP head was adopted following SimCLR. Note, that the MLP is only used during unsupervised training and is not intended for downstream tasks. In terms of additional augmentations, MoCo v2 also adopts the blur operation used in SimCLR.

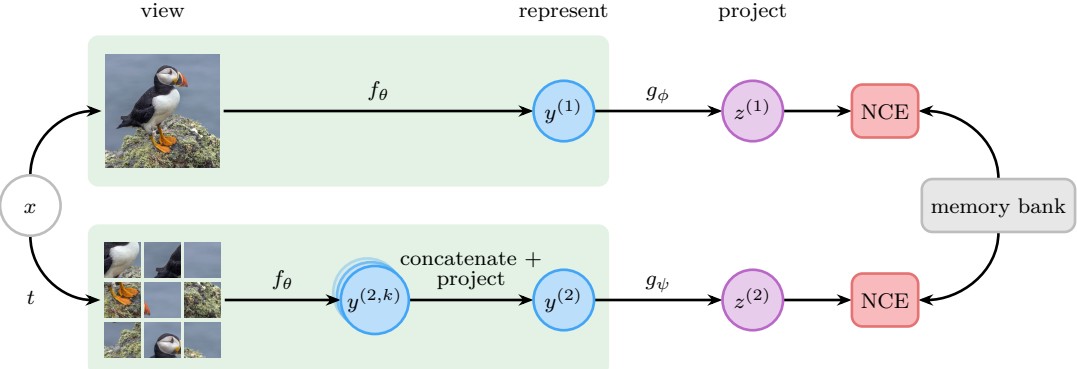

Figure 18: Architecture of PIRL. Minimizing a contrastive loss promotes similarity between the representations of the image and its corresponding transformation.

## 5.5 Pretext-Invariant Representation Learning (PIRL)

In the previously introduced pretext tasks we compute the representations of transformed images to predict properties from specific transformations, i.e., rotation angles (Noroozi & Favaro, 2016) or patch permutations (Gidaris et al., 2018). In this way, representations are encouraged to be covariant to the specific transformation, but are not guaranteed to capture the same underlying semantic information regardless of the transformation used. Although such covariance is advantageous in certain cases, we are more interested in representations that are semantically meaningful, so it is desirable to learn representations that are invariant to the transformation. In order to achieve this, Misra & Maaten (2020) refined the pretext task loss formulation and developed an approach called Pretext-Invariant Representation Learning (PIRL) which also makes use of memory banks (Wu et al., 2018).

The goal of PIRL is to train an encoder network $f_\theta$ that maps images $x_i^{(1)} = x_i$ and transformed images $x_i^{(2)} = t_\pi(x_i)$ to representations $y_i^{(1)}$ and $y_i^{(2)}$, respectively, which are invariant to the transformations used. Analogous to Section 2.3, $t_\pi$ denotes a jigsaw transformation consisting of a random permutation of image patches, where $\pi$ is the corresponding permutation. The loss formulation of pretext tasks as defined in Section 2 emphasizes that the encoder learns representations that contain information about the transformation rather than semantics. Let $z_i^{(1)} = g_\phi(f_\theta(x_i^{(1)}))$ and $z_i^{(2)} = g_\psi(f_\theta(x_i^{(2)}))$ be the projections obtained by the encoder $f_\theta$ and two separate projectors $g_\phi$ and $g_\psi$. The network is trained by minimizing a convex combination of two noise contrastive estimators (NCE) (Gutmann & Hyvärinen, 2010)

$$\mathcal{L}_{\theta,\phi,\psi}^{\text{PIRL}} = \frac{1}{n}\sum_{i=1}^{n} \lambda \ell_{\text{NCE}}\left(m_i, z_i^{(2)}, \bar{M}_i\right) + (1-\lambda)\ell_{\text{NCE}}\left(m_i, z_i^{(1)}, \bar{M}_i\right), \tag{59}$$

where $m_i$ is a projection from a memory bank corresponding to the original image $x_i$, each positive sample is assigned a randomly drawn set of negative projections $\bar{M}_i$ of images other than $x_i$ obtained from the memory bank, and $\lambda \in [0,1]$ is a hyperparameter. In contrast to the previously introduced pretext tasks, the loss formulation of PIRL does not explicitly aim to predict particular properties of the applied transformations, such as rotation or patch indices. Instead, it is solely defined on images and their corresponding transformed counterparts. NCE applies binary classification to each data point to distinguish positive and negative samples. Here, the NCE loss is formulated as

$$\ell_{\text{NCE}}(m, z, \bar{M}) = -\log[h(m, z, \bar{M})] - \sum_{\bar{m} \in \bar{M}} \log[1 - h(z, \bar{m}, \bar{M})], \tag{60}$$

where $h$ models the probability that $(x_i, x_i')$ is derived from $X$ as

$$h(u, v, \bar{M}) = \frac{\exp(s_{\cos}(u, v)/\tau)}{\exp(s_{\cos}(u, v)/\tau) + \sum_{\bar{m} \in \bar{M}} \exp(s_{\cos}(\bar{m}, v)/\tau)} \tag{61}$$

Table 1: Overview of the similarities and differences between the contrastive methods considered in this paper. The image transformations are either random, i.e., sampled from a predefined set $t \sim \mathcal{T}$, or deterministic. The number of positive pairs are counted per image.

| Method | CPC | CMC | SimCLR | MoCo | PIRL |
|---|---|---|---|---|---|
| Objective | InfoNCE | InfoNCE | InfoNCE | InfoNCE | NCE |
| Negative sampling | batch | memory bank | batch | queue | memory bank |
| Transformations | random + patches | deterministic | random | random | random (jigsaw) |
| Similarity | dot product | cosine sim. | cosine sim. | dot product | cosine sim. |
| # positive pairs | $mK$ | $m(m-1)$ | 2 | 1 | 2 |
| # encoders | 1 | $m$ | 1 | 2 | 1 |

for a temperature $\tau > 0$. Considering that the projections depend on the intermediate representations, the individual terms in Equation 60 encourage $y_i^{(1)}$ to be similar to $y_i^{(2)}$ and also $y_i^{(2)}$ to be dissimilar to the representations of other images. Since this formulation alone does not compare features between different untransformed images, the authors propose to use the convex combination of two NCE losses as defined in Equation 59. An overview of this approach is illustrated in Figure 18. The encoder network $f_\theta$ consists of the final layer of ResNet50 (He et al., 2016), average pooling and a 128-dimensional fully connected layer. As for the image transformation in the lower branch of Figure 18, we first extract nine image sections and apply them individually to $f_\theta$ to obtain patch representations $y_i^{(2,k)}$. These are then randomly concatenated and sent through another fully connected layer to obtain the 128-dimensional representation $y_i^{(2)}$. Although the authors focus their work on the Jigsaw pretext task, their approach can be generalized to any other pretext task. For demonstration purposes, the authors also conduct experiments with the rotation pretext task and its combination with the Jigsaw task. In this way we have to adapt the lower branch of Figure 18 by transforming the image at the beginning and feeding forward the transformed image to achieve the representation $y_i^{(2)}$ directly. Thus, using a secondary fully connected layer is not necessary anymore.

Note that PIRL can also be classified as a pretext task approach as defined in Section 2. However, it also uses ideas of contrastive representation learning which is why we decided to discuss it at this point.

### 5.6 Differences Between Contrastive Methods

In general, contrastive methods tend to blend into each other very quickly, where every one of them has its own twist. However, all methods in this paper share a similar objective, namely NCE (Gutmann & Hyvärinen, 2010) or InfoNCE (van den Oord et al., 2018), which essentially characterizes them as contrastive methods in a broader context. To highlight their differences, we summarize the design choices in Table 1.

### 5.7 Relations to Other Methods

**Pretext task methods.** Although contrastive learning is in itself sufficient to achieve state-of-the-art performance on downstream tasks, it is notable that Huang et al. (2022) are able to demonstrate a synergy between pretext tasks and contrastive learning. They use a method from the domain of masked image modeling (MIM), known as Masked Autoencoders (MAE), and combine it with ideas from this section. They emphasize the challenges that arise due to various distinctions, but can overcome them with adjustments to input augmentations, training objectives, model architectures and more. Their efforts achieve slightly better performance.

**Information maximization methods.** As stated in the introduction of this section, the encoder is encouraged to extract information that is unique to the anchor and the positive sample. More formally, van den Oord et al. (2018), Poole et al. (2019), and Song & Ermon (2020) show that minimizing the InfoNCE maximizes a lower bound on the mutual information between $y^*$ and $x^+$, i.e.,

$$I(Y^*; X^+) \geq \mathbb{E} \left[ - \operatorname{InfoNCE}_{s_\psi}(Y^*, Y^+, \bar{Y}) \right]. \tag{62}$$

Note, that the InfoNCE itself is upper bounded by $\log(n)$ where $n-1$ is the number of negative samples, so the bound from Equation 62 will be loose, if $I(Y^*; X^+) > \log(n)$ (Poole et al., 2019). Increasing the number of negative samples tightens this bound, and typically leads to better performance in practice. In this light, methods such as SimCLR and MoCo maximize the mutual information between two views, bearing a close resemblance to information maximization methods, e.g., VICReg, which maximizes the information content of the representations given the input samples.

Garrido et al. (2022) claim that non-contrastive methods, such as VICReg, actually are contrastive as well but on a feature level rather than contrasting entire samples. They propose two classes, one being *sample-contrastive* (see CPC, CMC or SimCLR), which minimize a contrastive objective on a batch of projections $Z \in \mathbb{R}^{d \times n}$ by considering the non-diagonal entries of its Gram matrix $Z^\top Z$, i.e.,

$$\mathcal{L}_{\mathrm{c}} = \|Z^\top Z - \mathrm{diag}(Z^\top Z)\|_F^2, \tag{63}$$

and the other being *dimension-contrastive* (see VICReg, Barlow Twins, or WMSE), which includes methods that minimize the non-diagonal entries of the covariance matrix of the projections

$$\mathcal{L}_{\mathrm{nc}} = \|ZZ^\top - \mathrm{diag}(ZZ^\top)\|_F^2. \tag{64}$$

This means sample-contrastive methods try to make positive and negative representations orthogonal, while dimension-contrastive methods make the feature dimensions of the representations orthogonal. However, the authors show if $Z$ is doubly-normalized (row and column normalized) that the following equality holds:

$$\mathcal{L}_{\mathrm{nc}} + \sum_{j=1}^{d} \|Z_{j,\cdot}\|_2^4 = \mathcal{L}_{\mathrm{c}} + \sum_{i=1}^{n} \|Z_{\cdot,i}\|_2^4, \tag{65}$$

where $Z_{\cdot,i} = z_i$ is the representation of an input sample and $Z_{j,\cdot}$ is a vector of the same feature across a batch. Their theorem proves that sample-contrastive and dimension-contrastive methods minimize similar objectives, but differ in practice especially when additional loss functions, e.g., the invariance loss from VICReg (Bardes et al., 2021), are introduced. In that case, $\mathcal{L}_{\mathrm{c}}$ and $\mathcal{L}_{\mathrm{nc}}$ cannot be used interchangeably.

**Teacher-student methods.** Several recent works argue that teacher-student methods perform an implicit type of contrastive learning. Tao et al. (2022) claim that BYOL can be connected to contrastive methods, although it does not use negatives explicitly. Roughly, the weight matrix of BYOL's predictor network slowly approaches a feature correlation matrix of the representations, leading to gradients with similar effects as negative samples. Fetterman & Albrecht (2020); Tian et al. (2020) argue that methods such as BYOL implicitly perform contrastive learning via batch normalization. More specifically, the batch normalization layer redistributes the inputs according to mean and standard deviation. Thus, if before batch normalization different examples are distributed very similarly (indicating representation collapse), after batch normalization the outputs will be redistributed, hence preventing representation collapse. Taking into account Figure 9, batch normalization identifies the common mode between projections in $Z^{(1)}$, which is then compared against the projections in $Z^{(2)}$ which implicitly serve as negative samples. Thus, according to Fetterman & Albrecht (2020), BYOL learns by considering how a sample is different from the average of other samples. Note, that here the samples are the augmented images for teacher and student respectively. Due to architectural similarities, this also holds for SimSiam, where Chen & He (2021) show that batch normalization is crucial to the method's performance. Taking a look at DINO's architecture seems to confirm this implicit contrastive learning. While DINO does not use batch normalization, it has a centering term for the teacher as explained in Section 4.2. The goal is to center the output probability distribution of the teacher to prevent representation collapse. Here too, the centering value is calculated across the batch, which captures the common mode between samples in the mini-batch.

## 6 Clustering-Based Methods

So far, some of the presented representation learning methods define a classification problem with hand-crafted labels to solve an auxiliary task (see Section 2). Instead of specifying these class labels by hand, clustering algorithms, e.g., k-means Lloyd (1982), can be used to create the labels in an unsupervised fashion.

The objective of clustering-based representation learning is to group images with similar representations into clusters. In contrastive learning, for example, this would allow us to discriminate between cluster assignments rather than individual images or representations, which significantly increases efficiency. Over the time, numerous clustering-based methods have been developed, each with its own strengths and weaknesses. In the subsequent sections, we present the most significant approaches.

## 6.1 DeepCluster

The first approach that implements the idea of clustering for representation learning is DeepCluster (Caron et al., 2018), which alternates between inventing pseudo-labels via cluster assignments and adjusting the representation to classify images according to their invented labels. The motivation behind this is to increase the performance of convolutional architectures that already exhibit a strong inductive bias, as these already perform reasonably well with randomly initialized weights (Noroozi & Favaro, 2016). Overall, the authors propose to repeatedly alternate between the following two steps to further improve the encoder network:

1. Group the representations $y_i = f_\theta(x_i)$ produced by the current state of the encoder $f_\theta$ to $k$ clusters (e.g., by using $k$-means clustering).

2. Use the cluster assignments from step 1 as pseudo-labels $\beta_i$ for supervision to update the weights, i.e.,

$$\mathcal{L}_{\theta,\psi}^{\text{DeepCluster}} \frac{1}{n} \sum_{i=1}^{n} d_{\text{classification}}(q_\psi(y_i), \beta_i), \tag{66}$$

where a predictor network $q_\psi$ tries to predict the cluster assignments of the representations $y_i = f_\theta(x_i)$.

In their experiments, the authors utilize a standard AlexNet (Krizhevsky et al., 2017) with $k$-means and argue that the choice of the clustering algorithm is not crucial.

## 6.2 Self Labelling (SeLa)

A common weakness of the naive combination of clustering and representation learning is its proneness to degenerate solutions where, for example, all representations are assigned to the same cluster. To circumvent this issue, Asano et al. (2019) have developed an refined alternating update scheme, called Self Labelling (SeLa), which imposes constraints on the labels such that each cluster is assigned the same amount of data points. The pseudo-labels corresponding to images $x_1, \ldots, x_n$ are encoded as one-hot vectors $\beta_1, \ldots, \beta_n \in \{0, 1\}^k$. To assign the pseudo-labels $\beta_1, \ldots, \beta_n$ and fit encoder and predictor networks $f_\theta$ and $q_\psi$, respectively, the authors consider the optimization problem

$$\min_{\beta,\theta,\psi} \quad \frac{1}{n} \sum_{i=1}^{n} d_{\text{ce}}(q_\psi(f_\theta(x_i)), \beta_i), \tag{67}$$

$$\text{s.t.} \quad \sum_{i=1}^{n} \beta_i[j] = \frac{n}{k}, \quad \beta_i[j] \in \{0, 1\} \quad \text{for } j \in \{1, \ldots, k\}, \tag{68}$$

which they solve by alternating between the following two steps:

1. The problem of assigning the pseudo-label to the images is formulated as an optimal transport problem which is solved using a fast variant of the Sinkhorn-Knopp algorithm (Cuturi, 2013).

2. Fix the pseudo-labels from step 1 and update the parameters $\theta$ and $\psi$ by minimizing the cross-entropy loss.

Note that step 2 is the same as used in DeepCluster. However, due to the choice of $k$-means in DeepCluster it is quite possible to obtain underrepresented cluster assignments that would lead to trivial solutions, i.e., constant representations are learned and consequently the minimum is achieved in both optimization steps.

## 6.3 Swapping Assignments Between Multiple Views (SwAV)

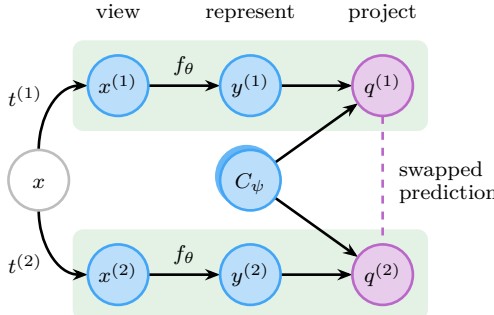

Figure 19: SwAV does not actually measure the similarity between image representations of different views, but instead compares the representations with codes obtained by assigning the features to parameterized prototypes.

In general, contrastive methods are computationally challenging due to the need for numerous explicit pairwise feature comparisons. However, Caron et al. (2020) propose an alternative algorithm, called SwAV, that circumvents this problem by data clustering while promoting consistency among cluster assignments across different views. In contrast to DeepCluster and SeLa, SwAV is an online clustering-based approach, i.e., it does not alternate between a cluster assignment and training step. An encoder network $f_\theta$ is used to compute image representations $y^{(1)}$ and $y^{(2)}$ of two views of the same image $x$. These representations are then mapped to a set of $k$ parameterized prototypes $C_\psi = [c_1, \ldots, c_k]$, resulting in corresponding codes $q^{(1)}$ and $q^{(2)}$. Next, a swapped prediction problem is addressed, where the codes derived from one view are predicted using the encoding from the second view. To achieve this, we minimize

$$\mathcal{L}_{\theta,\psi}^{\text{SwAV}} = \frac{1}{n} \sum_{i=1}^{n} \ell(q_i^{(1)}, y_i^{(2)}) + \ell(q_i^{(2)}, y_i^{(1)}), \tag{69}$$

where $\ell(q, y) = d_{\text{ce}}(q, \text{softmax}_\tau(C^\top y))$ quantifies the correspondence between the representation $y$ and the code $q$ for a temperature $\tau > 0$. For an overview of the architecture we refer to Figure 19. Note that, although SwAV takes advantage of contrastive learning, it does not require the use of a large memory bank or a momentum network.

In addition to this method, the authors also propose the augmentation technique called *multi-crop*, which was also used for DINO (see Section 4.2). Instead of using two views with full resolution, a mixture of views with different resolutions is used. In this approach, multiple transformations are compared by using considerably smaller ones, which leads to a further improvement of previous methods such as SimCLR, DeepCluster and SeLa.

## 6.4 Relations to Other Methods

The idea of clustering-based representation learning is to exploit the coherence between image representations and its corresponding cluster assignments. We are interested in learning image representations which get assigned to the same cluster if the underlying images are semantically similar.

Other self-supervised methods, such as pretext task, information maximization or teacher-student methods build upon loss formulations that consider images and their transformations as separate classes. In contrast, clustering-based methods like DeepCluster and SeLa discriminate between pseudo-label assignments of similar representations instead of individual images. The only exception could be DINO (Caron et al., 2021), which we have mainly classified as a teacher-student method. On closer inspection, we can see that it also has some similarities with clustering-based methods. Here, the last layer of the projector can be considered as cluster prototypes, which act as subtle clustering of the output of the first part of the projector (Garrido et al., 2022).

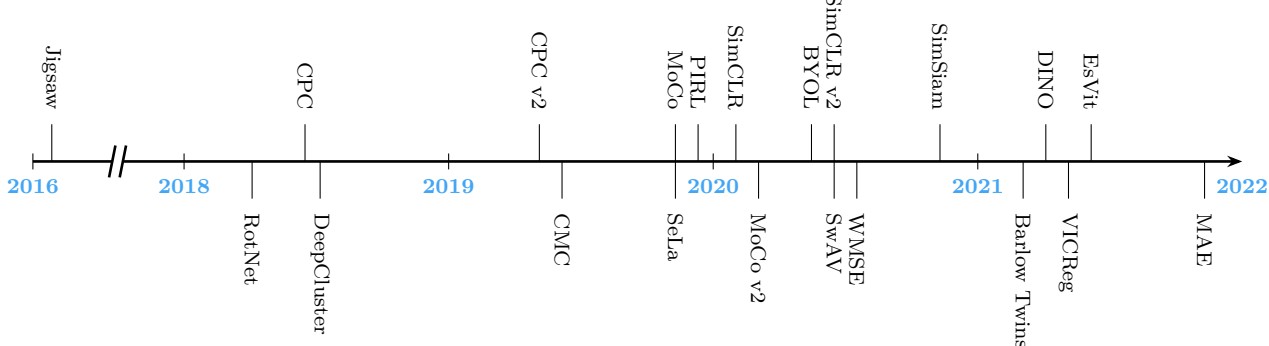

Figure 20: Timeline of the first date of publication on arXiv for every method.

The introduction of contrastive losses and memory banks also deviate from the notion of instance classes. However, these methods typically rely on a large number of pairwise feature comparisons, whereas linking with clustering-based methods, such as those done by Caron et al. (2020) in SwAV, can be much more memory efficient and can result in better performance.

## 7 Taxonomy of Representation Learning Methods

As we have seen in this survey, there are several ways to learn meaningful representations of images. These include solving specific tasks for pre-training such as predicting the rotation angle of an image, maximizing the mutual information between different views of the same image, using a contrastive loss in order to separate positive and negative samples in latent space, learning from a teacher network and clustering and subsequently self-labeling images. Based on these distinctions, we adapt and expand the taxonomy, proposed by Bardes et al. (2021) in the following section, which includes the following five categories:

1. Pretext task methods

2. Information maximization methods

3. Teacher-student methods

4. Contrastive methods

5. Clustering-based methods

Note that some methods fit into multiple categories, as they combine different approaches. CPC (v2) and CMC, e.g., both use a contrastive loss, as well as information maximization. PIRL includes solving a pretext and a contrastive loss to learn representations. We used this taxonomy to structure our paper, Figure 20 shows the chronological order in which all discussed methods have been published.

Figure 21 gives a visual overview on all methods within the proposed taxonomy. The inner nodes show the five categories that are connected to each reviewed method. Methods that can be assigned to several categories have multiple ingoing edges. As an additional overview, Table 2 lists all methods including their primary class assignment and the URL to the original implementation on Github[1], if available.

## 8 Meta-Study of Quantitative Results

Accessing the quality of the learned representations can be tricky. One approach that has become established in literature is to evaluate the obtained representations on downstream computer vision tasks, such as image

---

[1] https://github.com

Table 2: Overview over the representation learning approaches discussed in this paper.

| Method | Class | Code |
|---|---|---|
| Autoencoders (Le Cun, 1987) | Pretext-Task | |
| RotNet (Gidaris et al., 2018) | Pretext-Task | https://github.com/gidariss/FeatureLearningRotNet |
| Jigsaw (Noroozi & Favaro, 2016) | Pretext-Task | https://github.com/MehdiNoroozi/JigsawPuzzleSolver |
| MAE (He et al., 2022) | Pretext-Task | https://github.com/facebookresearch/mae |
| DINO (Caron et al., 2021) | Teacher-Student | https://github.com/facebookresearch/dino |
| EsViT (Li et al., 2021) | Teacher-Student | https://github.com/microsoft/esvit |
| BYOL (Grill et al., 2020) | Teacher-Student | https://github.com/deepmind/deepmind-research/tree/master/byol |
| VICReg (Bardes et al., 2021) | Info-Max | https://github.com/facebookresearch/vicreg |
| Barlow Twins (Zbontar et al., 2021) | Info-Max | https://github.com/facebookresearch/barlowtwins |
| SimSiam (Chen & He, 2021) | Info-Max | https://github.com/facebookresearch/simsiam |
| WMSE (Ermolov et al., 2021) | Info-Max | https://github.com/htdt/self-supervised |
| SimCLR (Chen et al., 2020a) | Contrastive | https://github.com/google-research/simclr |
| MoCo (He et al., 2020) | Contrastive | https://github.com/facebookresearch/moco |
| MoCo v2 (Chen et al., 2020b) | Contrastive | https://github.com/facebookresearch/moco |
| CPC (van den Oord et al., 2018) | Contrastive/Info-Max | |
| CPC v2 (Henaff, 2020) | Contrastive/Info-Max | |
| CMC (Tian et al., 2020) | Contrastive/Info-Max | https://github.com/HobbitLong/CMC |
| PIRL (Misra & Maaten, 2020) | Contrastive/Pretext-Task | https://github.com/facebookresearch/vissl/tree/main/projects/PIRL |
| DeepCluster (Caron et al., 2018) | Clustering | https://github.com/facebookresearch/deepcluster |
| SeLa (Asano et al., 2019) | Clustering | https://github.com/yukimasano/self-label |
| SwAV (Caron et al., 2020) | Clustering | https://github.com/facebookresearch/swav |

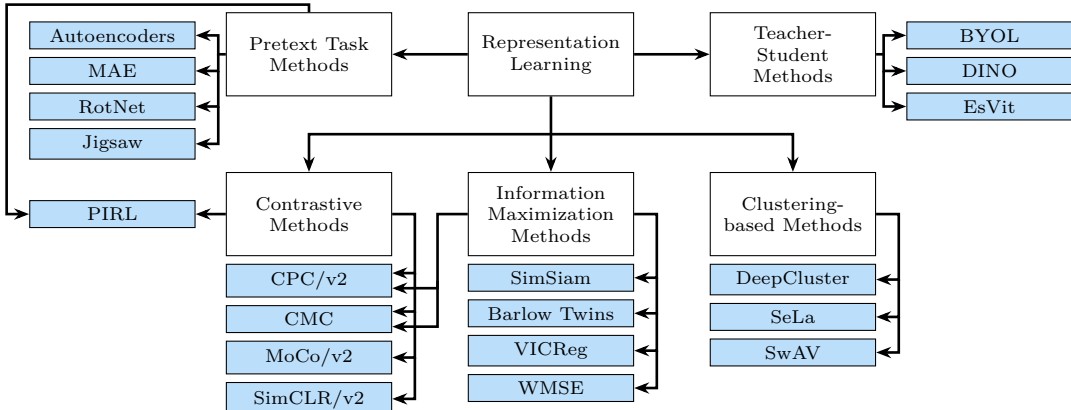

Figure 21: Graphical overview of the taxonomy of image representation learning methods.

classification, object detection, or instance segmentation. In this section we explain the evaluation process for representation learning methods and take a closer look at the most common evaluation tasks in literature. We compare all reviewed methods with regard to their performance and summarize the results for three different datasets. For experimental details please consider the original works. We perform a quantitative comparison and provide some insights on potential future directions to further evaluate and compare the reviewed methods.

## 8.1 Evaluation of Representation Learning Methods

The performance of representation learning models is often measured and compared by letting pre-trained models solve downstream tasks such as image classification. For pre-training, a base architecture is chosen as encoder and trained in a self-supervised manner without the use of labels. Many authors experiment with multiple architectures. One default architecture for image classification is the ResNet-50 (He et al., 2016), newer methods often use Vision Transformers (ViT) (Dosovitskiy et al., 2020) as encoder. The learned representations are then evaluated on different downstream tasks. We give more details on the evaluation protocols later in this section.

We identified five datasets that were most commonly used to evaluate representation learning methods: ImageNet (Russakovsky et al., 2015), the Pascal visual object classes (VOC) (Everingham et al., 2009), Microsoft common objects in context (COCO) (Lin et al., 2014), CIFAR-10, CIFAR-100 (Krizhevsky et al., 2009) and Places205 (Zhou et al., 2014). All listed datasets include one or multiple of the following tasks: image classification (IC), object detection (OD) and instance segmentation (Seg). Table 3 shows which of the most common datasets are used for evaluating each method.

To get a sense of the performance of all reviewed methods at a glance, we conduct a quantitative comparison in the following. We report the evaluation results for ImageNet, Pascal VOC and Microsoft COCO and identify gaps where further evaluation would be interesting. All methods have been pre-trained on the ImageNet training set before further evaluation.

**ImageNet.** Image classification on ImageNet, which includes 1000 different classes and has established as an evaluation standard for representation learning methods, is mostly evaluated in two ways. A linear classifier is either trained on top of the frozen pre-trained representations, or the model weights are initialized with the pre-trained weights and fine-tuned on 1% and 10% of the labeled ImageNet training data, respectively. Table 4 shows the accuracy for each method either using a ResNet-50 encoder for better comparability or a different architecture. We report both the top-1, as well as the top-5 accuracies. Among the methods evaluated with a ResNet-50, DINO and SwAV perform best, nearly reaching the performance of a supervised trained ResNet50. Considering the reported top-5 accuracy, BYOL performs best, right before VICReg and Barlow Twins. For other, bigger architectures, EsViT and DINO both perform best, while utilizing architectures with a comparatively little number of parameters.

Table 3: Overview on datasets (ImageNet, Pascal VOC, Microsoft COCO, CIFAR and Places-205) and tasks (image classification, object detection and instance segmentation) each representation learning method has been evaluated on. The numbers underneath indicate how many of the shown 20 methods used the corresponding dataset for evaluation.

| | ImageNet | VOC IC | VOC OD | VOC Seg | COCO OD | COCO Seg | CIFAR 10 | CIFAR 100 | Places 205 |
|---|---|---|---|---|---|---|---|---|---|
| RotNet | ✓ | ✓ | ✓ | ✓ | | | ✓ | | ✓ |
| Jigsaw | ✓ | ✓ | ✓ | ✓ | | | | | |
| DINO | ✓ | | | | | | ✓ | ✓ | |
| MAE | ✓ | | | | ✓ | ✓ | | | ✓ |
| EsViT | ✓ | ✓ | | | ✓ | ✓ | ✓ | ✓ | |
| BYOL | ✓ | ✓ | ✓ | ✓ | | | ✓ | ✓ | |
| VICReg | ✓ | ✓ | ✓ | | ✓ | ✓ | | | ✓ |
| Barlow Twins | ✓ | ✓ | ✓ | | ✓ | ✓ | | | |
| SimSiam | ✓ | | ✓ | | ✓ | ✓ | | | |
| WMSE | ✓ | | | | | | ✓ | ✓ | |
| SimCLR | ✓ | ✓ | | | | | ✓ | ✓ | ✓ |
| MoCo | ✓ | | ✓ | | ✓ | ✓ | | | |
| MoCo v2 | ✓ | | ✓ | | | | | | |
| CPC | ✓ | | | | | | | | |
| CPC v2 | ✓ | | ✓ | | | | | | |
| CMC | ✓ | | | | | | | | |
| PIRL | ✓ | ✓ | ✓ | | | | | | ✓ |
| DeepCluster | ✓ | ✓ | ✓ | ✓ | | | | | ✓ |
| SeLa | ✓ | ✓ | ✓ | ✓ | | | ✓ | ✓ | ✓ |
| SwAV | ✓ | ✓ | ✓ | | ✓ | ✓ | | | ✓ |
| # Total | 20 | 11 | 13 | 5 | 7 | 7 | 8 | 7 | 8 |

Table 4: Top-1 and top-5 accuracy on ImageNet classification for a linear evaluation (on the left) and semi-supervised learning, where the classifier is fine-tuned on 1% and 10% respectively of the labeled ImageNet data (on the right). The upper part shows performance for a ResNet-50 encoder and the lower part shows more results where other architectures have been used. We also report the number of parameters for each network.

| Method | #Params | Top-1 | Top-5 | Top-1 1% | Top-1 10% | Top-5 1% | Top-5 10% |
|---|---|---|---|---|---|---|---|
| *ResNet-50* | | | | | | | |
| Supervised (Zbontar et al.) | 24M | 76.5 | - | 25.4 | 56.4 | 48.4 | 80.4 |
| DINO | 24M | **75.3** | - | - | - | - | - |
| BYOL | 24M | 74.3 | **91.6** | 53.2 | 68.8 | 78.4 | 89.0 |
| VICReg | 24M | 73.2 | 91.1 | 54.8 | 69.5 | 79.4 | 89.5 |
| Barlow Twins | 24M | 73.2 | 91.0 | **55.0** | 69.7 | **79.2** | 89.3 |
| SimSiam | 24M | 71.3 | - | - | - | - | - |
| SimCLR | 24M | 69.3 | 89.0 | 48.3 | 65.6 | 75.5 | 87.8 |
| MoCo | 24M | 60.6 | - | - | - | - | - |
| MoCo v2 | 24M | 71.1 | 90.1 | - | - | - | - |
| CPC v2 | 24M | 63.8 | 85.3 | - | - | - | - |
| CMC[3] | 24M | 66.2 | 87.0 | | | | |
| PIRL | 24M | 63.6 | - | 30.7 | 60.4 | 57.2 | 83.8 |
| SeLa | 24M | 61.5 | 84.0 | | | | |
| SwAV with multi-crop | 24M | **75.3** | - | 53.9 | **70.2** | 78.5 | **89.9** |
| *other architectures* | | | | | | | |
| RotNet[4] (AlexNet) | | 55.4 | 77.9 | - | - | - | - |
| Jigsaw[4] (AlexNet) | | 44.6 | 68.0 | - | - | - | - |
| MAE (ViT-H) | 643M | 76.6 | - | - | - | - | - |
| DINO (ViT-B/8) | 85M | 80.1 | - | - | - | - | |
| EsViT (Swin-B) | 87M | **80.4** | - | - | - | - | - |
| BYOL (ResNet-200x2) | 250M | 79.6 | **94.8** | **71.2** | **77.7** | **89.5** | **93.7** |
| WMSE (ResNet-18) | | 79.0 | 94.5 | - | - | - | - |
| SimCLR (ResNet-50x4) | 375M | 76.5 | 93.2 | - | - | 85.8 | 92.6 |
| MoCo (ResNet-50x4) | 375M | 68.6 | - | - | - | - | - |
| CPC (ResNetv2-101) | 28M | 48.7 | 73.6 | - | - | - | - |
| CPC v2 (ResNet-161) | 305M | 71.5 | 90.1 | - | - | - | - |
| CPC v2 (ResNet-33) | | - | - | 52.7 | 73.1 | 78.3 | 91.2 |
| CMC[3] (ResNet-50x2) | 188M | 70.6 | 89.7 | - | - | - | - |
| DeepCluster (AlexNet) | 62M | 41.0 | - | - | - | - | - |

---

[3]trained with RandAugment (Cubuk et al., 2020).
[4]reported numbers are from Kolesnikov et al. (2019).

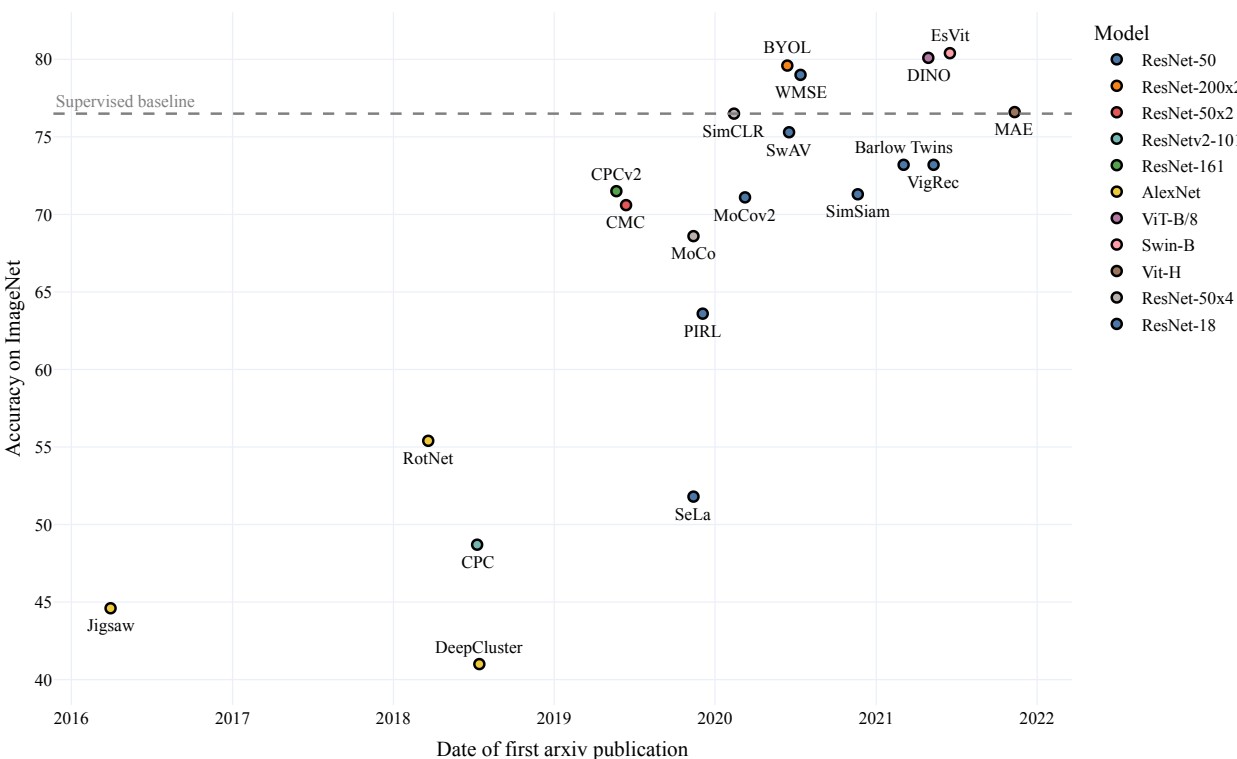

Figure 22: Reviewed methods Top-1 accuracy on ImageNet, sorted by their first publication on arXiv[2]. The gray line marks the supervised benchmark.

Figure 22 shows the best reported accuracy of all methods on the ImageNet training set in chronological order of their first publication on arXiv[2]. It would be interesting to measure the performance of every method under the same conditions to make them comparable. Nevertheless, the plot reveals some interesting points, e.g., that CPC v2 and CMC perform better than most other early published self-supervised methods and SwAV has not been beaten by any of the compared methods.

The goal of every representation learning method is to extract meaningful features from images that are useful for various tasks. The quality of extracted features learned on the ImageNet data can therefore be further assessed by transferring them to solve tasks on other datasets like the Pascal VOC and COCO object detection and instance segmentation. To evaluate the features on other tasks, the learned weights serve as initialization for a network and a linear classifier is trained on top, while the network layers are fine-tuned. Usually, the features are extracted from different layers of the network, while freezing the weights of the others. The best values are reported for each method. In the following we take a closer look at the results on the Pascal VOC and Microsoft COCO datasets. The goal when performing object detection on these two datasets is to predict bounding boxes for every object shown on the image. The task of instance segmentation is conducted pixel-wise, where every pixel is classified separately.

**Pascal VOC.** With only 20 classes, the Pascal VOC (Everingham et al., 2009) dataset is rather small and designed to resemble a real world setting. The data includes annotations for image classification, object detection, as well as instance segmentation. The standard metrics for the classification and object detection task are mean average precision (AP) with different intersection over union (IoU) threshold values. For the task of segmentation the mean IoU is reported. For a detailed overview on the aforementioned metrics we refer to the work of Padilla et al. (2020).

The first five columns of Table 5 show the results for the Pascal VOC tasks side by side for different architectures. For the object detection task Fast R-CNN (Girshick, 2015) and Faster R-CNN (Ren et al.,

---

[2]https://arxiv.org/

Table 5: Evaluation of a linear classifier on the PASCAL VOC image classification (IC), object detection (OD) and instance segmentation (Seg) and COCO OD and Seg tasks. For IC we report the mean average precision (mAP), the $AP_{all}$, $AP_{50}$ and $AP_{75}$ for the VOC OD, COCO OD and COCO Seg and for PASCAL VOC Seg the mean intersection over union (mIoU). We report the best values reported in the original papers using different architectures.

| Method | IC | Pascal VOC | | | | COCO | | | | | |
| | AP | OD $AP_{all}$ | $AP_{50}$ | $AP_{75}$ | Seg mIoU | OD $AP_{all}^{BB}$ | $AP_{50}^{BB}$ | $AP_{75}^{BB}$ | Seg $AP_{all}^{MK}$ | $AP_{50}^{MK}$ | $AP_{75}^{MK}$ |
|---|---|---|---|---|---|---|---|---|---|---|---|
| RotNet[5] | 72.9 | 54.4 | - | - | 39.1 | - | - | - | - | - | - |
| Jigsaw[5] | 67.6 | 53.2 | - | - | 37.6 | - | - | - | - | - | - |
| MAE | - | - | - | - | - | **53.3** | - | - | **47.2** | - | - |
| EsViT | 85.5 | - | - | - | - | 46.2 | **68.0** | **50.6** | 41.6 | **64.9** | **44.8** |
| BYOL | 85.4 | - | 77.5 | - | **76.3** | - | - | - | - | - | - |
| VICReg | 86.6 | - | 82.4 | - | - | 39.4 | - | - | 36.4 | - | - |
| Barlow Twins | 86.2 | 56.8 | **82.6** | 63.4 | - | 39.2 | 59.0 | 42.5 | 34.3 | 56.0 | 36.5 |
| SimSiam | - | 57.0 | 82.4 | 63.7 | - | 39.2 | 59.3 | 42.1 | 34.4 | 56.0 | 36.7 |
| SimCLR | 80.5 | - | - | - | - | - | - | - | - | - | - |
| MoCo | - | 55.9 | 81.5 | 62.6 | - | 40.8 | 61.6 | 44.7 | 36.9 | 58.4 | 39.7 |
| MoCo v2 | - | 57.4 | 82.5 | **64.0** | - | - | - | - | - | - | - |
| CPC v2 | - | 76.6 | - | - | - | - | - | - | - | - | - |
| PIRL | 81.1 | 54.0 | 80.7 | 59.7 | - | - | - | - | - | - | - |
| DeepCluster | 73.7 | 65.9 | - | - | 45.1 | - | - | - | - | - | - |
| SeLa | 75.9 | **57.8** | - | - | 44.7 | - | - | - | - | - | - |
| SwAV with multi-crop | **88.9** | 56.1 | **82.6** | 62.7 | - | 41.6 | 62.3 | 45.5 | 37.8 | 59.6 | 40.5 |

[5] uses a data dependent initialization as proposed by Krähenbühl et al. (2015).

2015) are most widely used. SwAV with multicrop performs best on the image classification task, just before VICReg and Barlow Twins. Barlow Twins and SwAV also perform best on the object detection task. From all methods that have been evaluated on the segmentation task, BYOL, by far, outperforms every other method. However, the results cannot be deemed as representative due to a lack of comparative values for other methods.

**COCO.** The Microsoft COCO dataset is a large dataset for object detection and segmentation including objects from 91 different classes, depicted in their natural setting. In the second half of Table 5 we present the average precision for object detection and instance segmentation. In the case of object detection it is the bounding box (BB) AP and in case of the segmentation it is the AP of the segmentation mask (MK). Note that we again report values for different encoder architectures. The most commonly used architecture for both object detection and segmentation is a Mask R-CNN (He et al., 2017) model with either the C4 or feature pyramid network (Lin et al., 2017) encoder. For both tasks the overall AP is best for MAE, other values are missing and EsViT outperforms every other method for the AP50 and AP75. Again, for a profound insight a more detailed experimental evaluation is necessary.

### 8.2 Future Directions

To conclude our quantitative meta-study we want to point out some interesting insights and suggest some experiments to conduct in the future.

As the comparison of all representation learning methods has shown, the performance strongly depends on the used network architecture. Some architectures, datasets and tasks have been utilized throughout various works to evaluate the quality of self-supervised trained image representations. Nevertheless, there is no standardized benchmark for a consistent comparison of methods, yet. Goyal et al. (2019) suggest a range of tasks as benchmark to cover many aspects and obtain a detailed comparison of methods in the future.

One main contribution of this meta-study is the categorization and comprehensive presentation of different approaches to the overall aim to learn meaningful representations of images. In Section 7 five main categories have been described into which different approaches can be grouped. There already are some methods that combine multiple approaches that work well, which shows that the combination of different representation learning methods holds potential for future research.

## 9 Conclusion

The goal of self-supervised representation learning is to extract meaningful features from unlabeled image data and use them to solve all kinds of downstream tasks. In this work we saw different strategies for representation learning and how they are related. We gave an extensive overview of methods that have been developed over the last years and adapted a framework to categorize them.

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
