# OpenReview forum: "A Survey on Self-Supervised Representation Learning"
_TMLR — Rejected by TMLR_

### Review · Reviewer_AMgD · 2023-10-01

**Summary Of Contributions:**

This paper reviewed the recent advances in self-supervised representation learning. The authors revisited the typical works of the pretext task based method, the information max based method, teacher-student based method, the contrastive based method, and the clustering based method. They also provided experimental comparisons for their reviewed methods on classification and detection datasets.

**Audience:**

Yes

**Claims And Evidence:**

Yes

**Requested Changes:**

See the above.

**Strengths And Weaknesses:**

The paper is well organized and clearly written. I really appreciate Fig. 20, which gives a quite clear overview of different types of methods. Here are just some suggestions:

1). The in-depth relationship between different types of methods can be further discussed. Nowadays, it is usually easy for us to understand the relationship among the same type of methods, but there are also lots of connections for different types of methods. For example, there are some works that reveal the mathematical equivalence between (masked)Autoencoder (i.e., the 1st type) and contrastive learning (i.e., the 4th type). There are also some works that utilize the clustering techniques (i.e., the 5th type) to deal with the biased issue in contrastive learning (e.g., SwAV). Actually, those different types of methods are not really independent of each other. So it would be nice if we could have more discussion among different methods.

2). Theoretical works should also be reviewed. As TMLR is a machine-learning journal, I would be very happy to see some theoretical foundations of self-supervised learning in this draft. At least, there are already lots of theoretical analyses for the effectiveness of Autoencoder and contrastive learning. The authors could consider adding them in some new sections.

3). Details of experimental settings should be further clarified. The experiment part of the current draft is quite weak, and lots of critical details are not clearly discussed. For example, we usually have very different settings for MAE and contrastive learning in our common implementations. How can we really make them fair comparisons on classification or detection datasets? Moreover,  I also encourage the authors to further discuss the advantages and disadvantages of different types of methods in experimental evaluations (e.g., the memory overhead, the training time, and the generalizability).

In general, I think this is a good paper and I would like to vote for a minor revision for the current draft.

---

> ### Author Response · Authors · 2023-10-30
> **Author's Response to Reviewer AMgD**
>
> > Strengths and Weaknesses:
>
> > The paper is well organized and clearly written. I really appreciate Fig. 20, which gives a quite clear overview of different types of methods.
>
> We are very glad that the reviewer appreciates the effort we put into this submission!
>
> > Here are just some suggestions:
>
> > 1). The in-depth relationship between different types of methods can be further discussed. Nowadays, it is usually easy for us to understand the relationship among the same type of methods, but there are also lots of connections for different types of methods. For example, there are some works that reveal the mathematical equivalence between (masked)Autoencoder (i.e., the 1st type) and contrastive learning (i.e., the 4th type). There are also some works that utilize the clustering techniques (i.e., the 5th type) to deal with the biased issue in contrastive learning (e.g., SwAV). Actually, those different types of methods are not really independent of each other. So it would be nice if we could have more discussion among different methods.
>
> We included such a discussion now in the new version of our paper. Please, refer to Sections 3.4, 4.5, 5.7, 6.4 and also the general response.
>
> > 2). Theoretical works should also be reviewed. As TMLR is a machine-learning journal, I would be very happy to see some theoretical foundations of self-supervised learning in this draft. At least, there are already lots of theoretical analyses for the effectiveness of Autoencoder and contrastive learning. The authors could consider adding them in some new sections.
>
> Thank your for the suggestion. We added Subsection 5.7 in which we discuss the theoretical connection and similarities between contrastive methods and information maximization methods.
>
> > 3). Details of experimental settings should be further clarified. The experiment part of the current draft is quite weak, and lots of critical details are not clearly discussed. For example, we usually have very different settings for MAE and contrastive learning in our common implementations. How can we really make them fair comparisons on classification or detection datasets? Moreover, I also encourage the authors to further discuss the advantages and disadvantages of different types of methods in experimental evaluations (e.g., the memory overhead, the training time, and the generalizability).
>
> In Section 8 we summarize the results of the experiments that have been performed in the original papers. We want to stress that we did not perform these experiments by ourselves but rather collected the experimental results to give an overview of the performance of each method. We changed the wording in the introductory text of Chapter 8 to make this more clear to the reader.
>
> > In general, I think this is a good paper and I would like to vote for a minor revision for the current draft.
>
> Thank you for your valuable feedback! We hope that we could further improve our draft by incorporating your feedback!

---

### Review · Reviewer_mwJF · 2023-10-04

**Summary Of Contributions:**

The paper is more like a simple grouping of existing methods, but lacking more in-depth thinking. I expect the authors to provide more understandings on the development of unsupervised representation learning methods. By summarizing the previous works, can we arrive at some conclusions to guide the development of representation learning? The current version of paper could bring few insights to the community and a through improvement is needed.

**Audience:**

Yes

**Claims And Evidence:**

Yes

**Requested Changes:**

1. Since Y usually corresponds to the label, I recommend using H to represent the feature/representation.
2. Why BarlowTwins and VICReg belong to information maximization methods? I feel they are more based on feature regularization.
3. I think it would be better to introduce different methods over time. For example, contrastive learning methods are proposed before Teacher-student and MAE, but the authors introduce them reversely.
4. Instead of simply listing different methods, it is important to make comparisons between different ones. For example, what is the relationship between CPC, CMC, and SimCLR? What are in common? What are different?
5. Why in DeepCluster it is possible for all data points to be grouped into a single cluster?
6. The paper is more like a simple grouping of existing methods but lacks more in-depth thinking. I expect the authors to provide more understanding on the development of unsupervised representation learning methods. By summarizing the previous works, can we arrive at some conclusions to guide the development of representation learning? The current version of the paper could bring a few insights to the community and a thorough improvement is needed.

**Strengths And Weaknesses:**

1. This paper reviews existing representation learning methods and paradigms, as well as datasets used for benchmarking.
2. Each pretext task is intuitively and clearly illustrated for better understanding.

---

> ### Author Response · Authors · 2023-10-30
> **Author's Response to Reviewer mwJF**
>
> > Strengths:
> > - This paper reviews existing representation learning methods and paradigms, as well as datasets used for benchmarking.
> > - Each pretext task is intuitively and clearly illustrated for better understanding.
> >
> We are happy that you found our description intuitive and clear! We hope that we could further improve our work by including your valuable feedback.
>
> > Requested Changes:
> > - Since Y usually corresponds to the label, I recommend using H to represent the feature/representation.
>
> We agree that it is often the case that y is the label in a supervised setting. The reasoning behind our notation is that we use three consecutive letters x, y, and z for input, representation, and projection, respectively. By doing so, we want to stress the sequential nature of these three, i.e., input -> representation -> projection. We hope that we could convince the reviewer that this notation has some benefit over the proposed notation (x, h, z). If this is an acceptance blocker, we would reconsider our choice but for now we prefer our notation.
> > - Why BarlowTwins and VICReg belong to information maximization methods? I feel they are more based on feature regularization.
>
> We agree that these methods are based on feature regularization and occasionally classified as feature decorrelation (as a form of regularization) methods in the literature but view this regularization as a technique serving the overall intention of information maximization, which ultimately aligns with the goal of preventing information collapse. This is also in line with the stated objectives and intuitions in the publications of Barlow Twins and VICReg. To emphasize the relation between the terms, we updated the introduction in Section 3 accordingly.
>
> > - I think it would be better to introduce different methods over time. For example, contrastive learning methods are proposed before Teacher-student and MAE, but the authors introduce them reversely.
>
> Thank you for this suggestion! We agree that introducing the different sections in the order in which they were introduced in the literature makes sense. The reasoning behind our ordering is that we wanted to start with the pretext-task methods as they are the easiest to understand (as you also pointed out). We found these methods appropriate to gently introduce the reader to the topic. We hope that the intuition that we (hopefully) conveyed in the pretext-task section (Section 2) makes it easier to understand more complex methods like information-maximization methods (Section 3), teacher-student methods (Section 4), or contrastive methods (Section 5). Furthermore, we think that Figure 21 also nicely shows the order in which the methods were introduced in the literature.
> > - Instead of simply listing different methods, it is important to make comparisons between different ones. For example, what is the relationship between CPC, CMC, and SimCLR? What are in common? What are different?
>
> We extended Section 5 on contrastive method with an additional Subsection 5.6 to compare the discussed methods. We choose to build a table which highlights the important aspects of every method. It includes negative sampling strategies, selection of the similarity measure, number of positive pairs among others.
>
> > - Why in DeepCluster it is possible for all data points to be grouped into a single cluster?
>
> Achieving a reasonable solution in Eq. 61 already requires reasonable cluster assignments from the clustering approach in Step 1. Without avoiding underrepresented (or empty) cluster assignments, it is easy to see that assigning all data points to a single cluster would result in a trivial solution. To make this more clear, we have updated the formulation in the paper:
> "However, due to the choice of $k$-means in DeepCluster it is quite possible to obtain underrepresented cluster assignments that would lead to trivial solutions, i.e., constant representations are learned and consequently the minimum is achieved in both optimization steps."
>
> > - The paper is more like a simple grouping of existing methods but lacks more in-depth thinking. I expect the authors to provide more understanding on the development of unsupervised representation learning methods. By summarizing the previous works, can we arrive at some conclusions to guide the development of representation learning? The current version of the paper could bring a few insights to the community and a thorough improvement is needed.
>
> We did a literature review and included some works that discuss interesting connections between different methods. Please refer to the new Sections 3.4, 4.5, 5.7, 6.4.

---

### Review · Reviewer_Fy5y · 2023-10-16

**Summary Of Contributions:**

The authors have written a comprehensive survey of the prominent representation learning methods published in the last few years. They group the methods into several large categories including pretext task methods, information maximization methods, teacher-student methods, contrastive learning methods, and clustering-based methods.  Their unified notations and graphical representation of the methods make it easy to read and serve as a good survey for new comers to the area. They also include a meta-study of experiments published in previous papers to compare the performance of the different algorithms.

**Audience:**

Yes

**Claims And Evidence:**

Yes

**Requested Changes:**

- In many figures such as Figures 6,7,8, the two branches use the same notation 't' as their transformation. This is potentially misleading since the two branches use different transformations sampled from the same distribution. It would be better to label them as t and t', for example, to indicate that they're sampled independently and can be different.

- Important concepts such as 'representation collapse' should be better defined. The authors try to define it in two sentences at the beginning of section 3 but refer frequently to it afterwards.

- I would like to see an expanded overview of the 5 main categories of methods, such as historically how these ideas are developed and their relations to one another. For example, in the pretext task section, the authors can discuss what the good choices of pretext asks are. In the information maximization section, the authors can illustrate what type of information are maximized for each of the methods. The pros and cons of these different categories of methods can also be discussed.

**Strengths And Weaknesses:**

Strengths:
- This survey paper is very clearly written and easy to read, with most of the methods succinctly described as appropriate for a survey. This can serve as an introduction to researchers new to the area, and I personally find it enjoyable to read.

- The unified mathematical notations and graphical representations make it easy to understand and appreciate the similarities and differences of the methods.


Weaknesses:
- The main weakness of current submission, though not an issue of the paper itself, is whether survey papers fall under the scope of TMLR under the current editorial policy (https://jmlr.org/tmlr/editorial-policies.html). The closest category in scope would be reproducibility studies. But the current paper does not run any new comparison experiments on the different methods. The meta-study only contains experimental results from previous published papers.

- Apart from the description of the individual methods and their categorization, there are relatively little discussions on the relations between the different big categories of methods or their historical developments. In many sections the explanations of the main ideas behind the big category of representation learning methods are very short and the authors go almost straight into the description of individual methods.

- The meta-study section is relatively weak because it only contains numbers from previous papers, and many papers published at a later date contained in this survey already have similar tables comparing their newly proposed method with previous methods. It would have more added value if it tries to study these published methods with specific comparisons, such as how their performance vary with the network backbone or choice of sets of input transformations.

---

> ### Author Response · Authors · 2023-10-30
> **Author's Response to Reviewer Fy5y**
>
> > Strengths
>
> > - This survey paper is very clearly written and easy to read, with most of the methods succinctly described as appropriate for a survey. This can serve as an introduction to researchers new to the area, and I personally find it enjoyable to read.
>
> > - The unified mathematical notations and graphical representations make it easy to understand and appreciate the similarities and differences of the methods.
>
> We are glad that the reviewer liked our paper! We hope that we can further strengthen this submission by including their constructive and very useful feedback!
>
> > Weaknesses:
>
> > - The main weakness of current submission, though not an issue of the paper itself, is whether survey papers fall under the scope of TMLR under the current editorial policy (https://jmlr.org/tmlr/editorial-policies.html). The closest category in scope would be reproducibility studies. But the current paper does not run any new comparison experiments on the different methods. The meta-study only contains experimental results from previous published papers.
>
> We think that surveys are within the scope of TMLR since (i) there are already a couple of survey published in TMLR (e.g., https://openreview.net/forum?id=YdMrdhGx9y, https://openreview.net/forum?id=lmXMXP74TO, https://openreview.net/forum?id=jh7wH2AzKK), and (ii) there is a "Survey Certification". For these reasons we decided to submit our survey to TMLR (although survey papers are not explicitly mentioned in the editorial policy, as you correctly noticed).
>
> > - Apart from the description of the individual methods and their categorization, there are relatively little discussions on the relations between the different big categories of methods or their historical developments. In many sections the explanations of the main ideas behind the big category of representation learning methods are very short and the authors go almost straight into the description of individual methods.
>
> We included this in the new version of our paper. Please, refer to Sections 3.4, 4.5, 5.7, 6.4 and also the general response.
>
> > - The meta-study section is relatively weak because it only contains numbers from previous papers, and many papers published at a later date contained in this survey already have similar tables comparing their newly proposed method with previous methods. It would have more added value if it tries to study these published methods with specific comparisons, such as how their performance vary with the network backbone or choice of sets of input transformations.
>
> We agree that the choice of network backbone and input transformations is an very interesting area to study. However, doing these experiments is beyond the scope of this survey paper as we only want to give an overview of the methods and set them in relation to each other. The aim of the meta-study was to collect the numbers from the original papers and give the reader a rough overview of the performance of these methods.
>
> > Requested Changes:
>
> > - In many figures such as Figures 6,7,8, the two branches use the same notation 't' as their transformation. This is potentially misleading since the two branches use different transformations sampled from the same distribution. It would be better to label them as t and t', for example, to indicate that they're sampled independently and can be different.
>
> Thank you for this suggestion! We incorporated it in the current version of our submission. To keep the notation consistent, we used $t^{(1)}$ and $t^{(2)}$ instead of $t$ and $t'$.
>
> > - Important concepts such as 'representation collapse' should be better defined. The authors try to define it in two sentences at the beginning of section 3 but refer frequently to it afterwards.
>
> Thank you for bringing this to our attention. We updated the definition of representation collapse by adding a practical definition, linked to the context of information theory, and outlining factors that facilitate representation collapse.
>
> > - I would like to see an expanded overview of the 5 main categories of methods, such as historically how these ideas are developed and their relations to one another. For example, in the pretext task section, the authors can discuss what the good choices of pretext asks are. In the information maximization section, the authors can illustrate what type of information are maximized for each of the methods. The pros and cons of these different categories of methods can also be discussed.
>
> As requested, we expanded the description of the method categories. Please, refer to Section 1.4, as well as, the first paragraphs of Section 3 and the new Sections 3.4, 4.5, 5.7, 6.4. We also added a timeline (Figure 20) to visualize in which ordering the methods were introduced in literature.

---

> ### Comment · Action_Editors · 2023-11-18
>
> Dear reviewer,
>
> While I can understand that you may be working on your own papers for some conferences and at the same time serving for many conferences/journals, could you please take a look at the rebuttal and submit your official recommendation? The deadline was Nov 14 and it is Nov 18 now.
>
> AE

---

> > ### Comment · Reviewer_Fy5y · 2023-11-18
> >
> > Dear AE,
> >
> > Sorry for the late response as I was traveling this week. I have submitted my official recommendation.
> >
> > Thanks!
> > Reviewer

---

### Author Response · Authors · 2023-10-30
**General Response**

First, we would like to thank everyone involved for the detailed reviews and thoughts on our submission. We have substantially revised our submission and hope that all raised concerns are adequately addressed.

All three reviewers mentioned that a discussions on the relationship between the different groups of methods was missing in the first submission. We now include Sections 3.4, 4.5, 5.7, 6.4 that discuss the connections between the different classes of methods.

We marked changes in the text and new sentences in red.

---

### Decision · Action_Editor_qtNw · 2023-11-20

**Recommendation:** Reject

**Comment:**

This is a reasonably good survey paper about self-supervised representation learning. After the rebuttal, one reviewer voted for acceptance and two voted for rejection. The positive reviewer said
> **Leaning Accept**: After going over the authors' replies and other reviewers' reviews, I believe the current paper is a good survey of the recent progress in self-supervised representation learning. Certain areas of the paper can be strengthened, such as further illuminating the relations between the methods, which the reviewers have all pointed out. The authors have addressed these issues in some way but not completely. But overall I think this is still a very useful survey paper beneficial to the community.

The two negative reviewers said
> **Leaning Reject**: My concerns about the review of theoretical works and the experimental details are not solved. It seems that the author did not really add the theoretical results of SSL, section 5.7 is still a relationship discussion.

> **Reject**: As presented in my initial review, the paper does not conduct a comprehensive survey and provide deep insights to the community. Although the authors made a revision which is however still lower than the TMLR bar.

I went through the submission by myself and found many issues that can be improved (see below) besides the issues pointed out by our reviewers. Thus, it needs at least a major revision, and we cannot accept its current version for publication.

-----

First of all, please take a closer look at the following QA from the TMLR FAQ.

> Q: Does TMLR accept survey papers?

> A: Yes. Authors should make sure to emphasize the contributions made by the survey. Ideally, we want survey papers that draw new, previously unreported connections between several pieces of work in an area, and/or that clearly highlight trends in the area and/or suggest currently open problems. It should also be noted that if a submission has more than 12 pages of main content, then TMLR's normal short review timeline will not be enforced.

As far as I know, this is not the first survey on self-supervised representation learning. Similar to a research paper, a survey paper must have its own motivation, namely, why we need this survey paper given the existence of previous surveys. In particular, the new survey should *draw new, previously unreported connections between several pieces of work in an area, and/or clearly highlight trends in the area and/or suggest currently open problems*. I don't think those are completely missing from the current submission, but I do think those are not sufficiently emphasized as the unique contributions of the current submission.

Specifically, I like Table 1 that conceptually gives the connections and differences of the methods in Section 5. I would like to see a bigger table covering all the methods in all the sections. The table can be in a higher level of abstraction, listing all the fundamental statistical/algorithmic components used in self-supervised representation learning, such as contrastive loss, pretext tasks, multiple views (data-augmentation paths), momentum encoder, and so on. It would be better to carefully explain these underlying components with great details, and then show the readers which method is a combination of which components in a shorter length. This research area is being developed so fast, and thus focusing more on the underlying components is more stable and more friendly than focusing on the combined methods.

-----

Next, please find some major issues that I identified when reading your submission.

**Title**: It might be too big. What has been surveyed is self-supervised representation learning for CV problems (as claimed by the authors), and the readers cannot know whether the surveyed methods can work well on domains other than natural images such as medical images and satellite images. The authors should be more specific for the title.

**Unsupported claims**: When introducing some methods, I saw some unsupported claims, for example,
> Without precaution the model could “cheat” by predicting image patches from neighboring pixels, since the information in natural images is usually very spatially redundant.
> To learn good representations, it is crucial that the masking ratio is very high, e.g., 75%, to encourage the encoder to extract more high-level features.

Since this is a survey paper rather than a research paper, every single (strong) claim should have an information source, for the readers to further investigate the related issues by themselves. The source can be either a research or survey paper as a reference, or the experiments done by the authors. In the latter case, please clearly indicate that you observed it and the corresponding experimental results can be found somewhere in this paper.

**Terminology**: Some terms sound quite strange (at least to me). For example,
> In some cases, the calculation of the representations can be decoupled such that each view is processed independently ... If this is possible we call the corresponding encoder a *Siamese* encoder.

No reference is given as the credit of the name. In fact, given the popularity of Siamese networks in the industry for information retrieval, question answering, and other similar applications, the introduction of the name Siamese encoder is very confusing. Again, this is a survey paper rather than a research paper, so that you expect more readers with more diverse backgrounds and I think it is best to follow the most popular terminology.

**Taxonomy**: Last but not least, it is unclear why the 5 categories were abstracted from the surveyed methods and how each method was assigned into one of these categories. I just give one example but please be aware that there are several such examples: "4.4 Simple Siamese Representation Learning (SimSiam)" is currently under "4 Teacher-Student Methods"; however, the authors mentioned that
> However, teacher and student share the same parameters and hence a momentum encoder is not used as in previously presented teacher-student methods.

> Recent work has shown that teacher-student methods seem to perform implicit contrastive learning.

I got confused that if the two views are entirely symmetric, why one is the teacher network and the other is the student network? Why 4.4 is not under Section 3 or 5?

I guess the authors have some good reasons for doing so (i.e., why the 5 categories were abstracted from the surveyed methods and how each method was assigned into one of these categories), but those reasons should be clearly given in the introduction. Together with the motivation of having a new survey, a detailed explanation about the taxonomy used to organize the surveyed methods is extremely important for the introduction of such a paper. Designing a new taxonomy can also be a unique contribution as long as why the new taxonomy is better has been clearly explained.

-----

Hope the above comments are constructive and helpful.

**Audience:**

Yes.

**Claims And Evidence:**

Yes.

**Resubmission Of Major Revision:**

The authors may consider submitting a major revision at a later time.